# Membrane particles from mesenchymal stromal cells reduce the expression of fibrotic markers on pulmonary cells

Ana Merino[1]☯*, Martin J. Hoogduijn[1], Maria Molina-Molina[2,3], Elena G. Arias-Salgado[4], Sander S. Korevaar[1], Carla C. Baan[1], Ana Montes-Worboys[2]☯*

**1** Department of Internal Medicine, Nephrology and Transplantation, Erasmus Medical Center, Rotterdam, The Netherlands, **2** Unit of Interstitial Lung Diseases, Pulmonary Department, University Hospital of Bellvitge, IDIBELL, Hospitalet de Llobregat, Barcelona, Spain, **3** CIBER of Respiratory Diseases (CIBERES) Health Institute Carlos III, Madrid, Spain, **4** Instituto de Investigaciones Biomédicas, CSIC/UAM, Madrid, Spain

☯ These authors contributed equally to this work.
* a.merinorodriguez@erasmusmc.nl (AM); amontesw@idibell.cat (AMW)

**Data Availability Statement:** All relevant data are within the manuscript and its Supporting Information files.

## Abstract

### Background

Idiopathic pulmonary fibrosis (IPF) is a devastating lung disease with limited treatment options in which the telomere shortening is a strong predictive factor of poor prognosis. Mesenchymal stromal cells (MSC) administration is probed in several experimental induced lung pathologies; however, MSC might stimulate fibrotic processes. A therapy that avoids MSC side effects of transformation would be an alternative to the use of living cells. Membranes particles (MP) are nanovesicles artificially generated from the membranes of MSC containing active enzymes involved in ECM regeneration. We aimed to investigate the antifibrotic role of MP derived from MSC in an *in vitro* model of pulmonary fibrosis.

### Methods

Epithelial cells (A549) and lung fibroblasts, from IPF patients with different telomere length, were co-cultured with MP and TGF-β for 48h and gene expression of major pro-fibrotic markers were analyzed.

### Results

About 90% of both types of cells effectively took up MP without cytotoxic effects. MP decreased the expression of profibrotic proteins such as Col1A1, Fibronectin and PAI-1, in A549 cells. In fibroblasts culture, there was a different response in the inhibitory effect of MP on some pro-fibrotic markers when comparing fibroblast from normal telomere length patients (FN) versus short telomere length (FS), but both types showed an inhibition of Col1A1, Tenascin-c, PAI-1 and MMP-1 gene expression after MP treatment.

**Funding:** This collaboration project is co-funded by the PPP Allowance made available by Health Holland, Top Sector Life Sciences & Health, of the Dutch Ministry of Economic affairs to stimulate public-private partnerships, and by the Institute of Health Carlos III (ISCIII, PI18/00367), co-funded by FEDER funds/European Regional Development Fund (ERDF) – "a Way to Build Europe" and Biomedical National Research Network CIBER.

**Competing interests:** The authors have declared that no competing interests exist.

## Conclusions

MP conserve some of the properties attributed to the living MSC. This study shows that MP target lung cells, via which they may have a broad anti-fibrotic effect.

## Introduction

Idiopathic pulmonary fibrosis (IPF) is the most common form of the idiopathic interstitial pneumonias. It is a chronic, progressive, irreversible, and lethal lung disease of unknown cause. IPF occurs in middle-aged and elderly adults [1–3] with a median survival of 2–4 years [2] after diagnosis. Telomere shortening is a strong predictive factor of poor prognosis [1, 4–6]; patients with IPF and short telomeres have more rapid disease progression [5]. The pathogenic mechanisms remain unclear, but a growing body of evidence indicates that the disease is the result of an abnormal behavior of the alveolar epithelial cells that provoke the migration, proliferation, and activation of mesenchymal cells, with the formation of fibroblast and myofibroblast foci [1]. It has been described that transforming growth factor (TGF)-β plays a crucial role as a pro-fibrotic cytokine [7]. Activated myofibroblasts secrete exaggerated amounts of extracellular matrix (ECM) molecules with the subsequent destruction of the lung architecture [1, 8], resulting in respiratory failure [9]. Although numerous therapies have been evaluated in patients with IPF [10], pirfenidone [11] and nintedanib [12] are the only treatments that slow down disease progression [13] but are not fully effective at reducing mortality. Thus, there is an urgent need to identify novel therapeutic targets for IPF [10] that will reverse or stop the progression of the disease [1, 3].

Accordingly, the use of stem cells to improve regeneration and reduce fibrosis has been reported in animal models of lung fibrosis [14–17]. Very recently, three phase 1 clinical trials demonstrated the feasibility and safety of administration of mesenchymal stromal cells (MSC) in IPF patients [18–20]. It is well established that MSC exert their therapeutic effect via secretion of bioactive products such as cytokines, growth factors, and extracellular vesicles (EV) [21]. EV derived from MSC have been found to promote therapeutic activities that are comparable to MSC themselves such as the suppression of pro-inflammatory processes and reduction of oxidative stress, pulmonary fibrosis and tissue remodeling in a variety of *in vivo* inflammatory lung disease models by transferring their components [22]. The major advantage of MSC-derived EV therapy over live-cell transplantation can be attributed to the inherent risks and limitations associated with live-cell transplants. However, the production of EV results a major challenge as, the available protocols to isolate EV do not avoid the co-precipitation of contaminant proteins non-associated to EV, such as soluble growth factor produced by the cells as it is described by the International Society of Extracellular Vesicles [23]. Furthermore, there is heterogeneity in EV molecular content due to the active process of EV release that depends on the different stimuli received by the origin cell [24]. Finally, the low yield of EV originated by the cells makes difficult a scalable production of these vesicles as therapeutic product [21, 22].

Recently, we have generated nanovesicles from the membranes of MSC named membrane particles (MP) [25]. This novel product shows a size smaller than 200 nm, and spherical shape, properties shared with EV. The main advantage of MP respect to another type of vesicles is the high yield of MP obtained per cell. The scalable production of MP makes them an attractive pharmaceutical product. We also showed that MP have comparable functional effects to MSC, such as immunomodulation [25], and endothelial regeneration [26]. These previous studies suggested that MP could have other therapeutic effects described for MSC such as their antifibrotic capacity.

In the current study, we used an *in vitro* fibrosis model based on the stimulation with TGF-β of bronchial alveolar human epithelial cells (A549) and fibroblasts from IPF patients with different length of telomere to investigate the anti-fibrotic role of MP derived from human adipose tissue mesenchymal stem cell (AT-MSC)

## Materials and methods

### Ethics statement and human tissue samples

Human AT-MSC were isolated from subcutaneous adipose tissue from healthy kidney donors available during kidney donation procedures. The tissues were collected after obtaining written informed consent, as approved by the Medical Ethical Committee of the Erasmus University Medical Centre Rotterdam (protocol no. MEC-2006-190).

### Isolation and culture of AT-MSC

AT-MSC were isolated from subcutaneous adipose tissue of four healthy donors (1 female/ 3males). The range of age of the donors was between 34–78 years old. The adipose tissue was mechanically disrupted and enzymatically digested with 0.5 mg/ml collagenase type IV (Sigma-Aldrich, St. Louis, MO) in RPMI medium for 30 min at 37˚C under continuous shaking. Thereafter, the cells were resuspended in MEM-$\alpha$ medium containing 10% inactivated fetal bovine serum (FBS) (Lonza, Verviers, Belgium), 2 mM L-glutamine and 1% penicillin/ streptomycin, filtered through a 100 μm cell strainer, and transferred to 175 cm$^2$ culture flasks (Greiner Bio-one, Essen, Germany). At 90% confluence AT-MSC (passage 2–6) were collected to generate cell MP.

### Generation and characterization of membrane particles from AT-MSC

As we described previously [25], AT-MSC were osmotically lysed by incubating them in ultra-pure H$_2$O at 4˚C until the cells liberated their nuclei. Cell extracts were cleared of unbroken cells and nuclei by centrifugation at 2,000 x *g* for 20 min. After discarding the nuclei, samples were transferred to Amicon Ultra-15 filter tubes (100 kDa pore size) and concentrated the crude membranes by centrifugation at 4,000 x *g* at 4˚C. The concentrated cell membranes were diluted in 0.2 μm filtered PBS.

A population of MP, with a size below 200 nm was obtained by extruding the cell membranes gradually through different pore size filters, first 800 nm, second 400 nm and third 200 nm to avoid the blockade of the smaller final pore size filter [27] (Merck, KGaA, Darmstadt, Germany).

The extrusion process was performed using LiposoFast LF-50 device (AVESTIN Europe, Mannheim, Germany) at a pressure of 20 psi to force the MP to pass through the filters. All procedures were performed on ice.

The preparations of MP were visualized by cryo-transmission electron microscopy (Cryo-TEM). A thin aqueous film was formed by applying a 3 μl droplet of MP suspension to a specimen bare EM grid. Glow-discharged holey carbon grids were used. After the application of the suspension, the grid was blotted against filter paper, leaving a thin sample film spanning the grid holes. These films were vitrified by plunging the grid into ethane, which was kept at its melting point by liquid nitrogen, using a Vitrobot (Thermo Fisher Scientific Company, Eindhoven, Netherlands) and keeping the sample before freezing at 95% humidity. The vitreous sample films were transferred to a Tecnai Arctica microscope (Thermo Fisher Scientific, Eindhoven, Netherlands). Images were taken at 200 Kv with a field emission gun using a Falcon III (Thermo Fisher Scientific) direct electron detector. The analysis of absolute size distribution

and concentration of MP was performed by Nanoparticle tracking analysis (NTA) using NanoSight NS300 (NanoSight Ltd.). The NTA measurement conditions were: detect threshold 3, three measurements per sample (30 s/measurement), temperature 23.61 ± 0.8˚C; viscosity 0.92 ± 0.02 cP, 25 frames per second. Each video was analyzed to obtain the mean, mode, median and estimated concentration for each particle size.

## Sample preparation and proteomic data analysis

MP were lysed in an ice-cold buffer containing 100 mM Tris-HCl (pH 8.5), 12 mM sodium DOC and 12 mM sodium N-lauroylsarcosinate. The lysate was sonicated for 10 min and boiled for 5 min at 95˚C. Proteins were subjected to reduction with dithiothreitol, alkylation with iodoacetamide and then in-solution digested with trypsin (sequencing grade. Promega). Proteolytic peptides were collected, washed and analyzed by liquid chromatography tandem mass spectrometry (nLC-MS/MS) performed on an EASY-nLC coupled to an Orbitrap Fusion Lumos Tribid mass spectrometer (Thermo) operating in positive mode. Peptides were separated on a ReproSil-C18 reversed-phase column (Dr Maisch; 15 cm × 50 µm) using a linear gradient of 0–80% acetonitrile (in 0.1% formic acid). Spectra were acquired in continuum mode; fragmentation of the peptides was performed in data-dependent mode by HCD.

Raw mass spectrometry data were analyzed with the MaxQuant software suite [28] with the additional options 'LFQ' and 'iBAQ' selected. The Andromeda search engine was used to search the MS/MS spectra against the Uniprot database taxonomy: *Homo sapiens*, (release: June 2019) concatenated with the reversed versions of all sequences. A maximum of two missed cleavages was allowed. The peptide tolerance was set to 10 ppm and the fragment ion tolerance was set to 0.6 Da for HCD spectra. The enzyme specificity was set to trypsin and cysteine carbamidomethylation was set as a fixed modification. Both the PSM and protein FDR were set to 0.01. In case the identified peptides of two proteins were the same or the identified peptides of one protein included all peptides of another protein, these proteins were combined by MaxQuant software suite and reported as one protein group. Before further statistical analysis, known contaminants and reverse hits were removed.

Proteomic data analysis Panther Pathway was used to detect the protein pathways involved in fibrosis in each sample and the number of proteins of each pathway represented in each sample (www.pantherdb.com).

## Matrix metalloproteinases (MMPs) activity

MMPs activity of MP was analyzed by fluorimetric assay (MMP Activity Assay Kit, ab112146; Abcam) following the recommended protocol. Two amounts of MP ($5x10^9$ MP and $2.5x10^9$ MP) were incubated with 2 mM 4-Aminophenylmercuric Acetate (APMA) during 3 h to activate the MMP. MP without APMA activation, and MP incubated with 10 µM of GM6001 (a pan-matrix metalloproteinase inhibitor) (ab120845; Abcam) after 3h with APMA were used as controls of the technique. The assay is based on a fluorescence resonance energy transfer (FRET) peptide as a generic MMP activity indicator. In the intact FRET peptide, the fluorescence of one part is quenched by another. After cleavage into two separate fragments by MMPs, the fluorescence is recovered. The signal was detected by a fluorescence microplate reader at Ex/Em = 490/525 nm.

## Lung epithelial cell line culture

Bronchial-alveolar epithelial human cells (A549) were purchased from the American Type Culture Collection (ATCC, Manassas, VA, USA) and cultured in F12K (Gibco Life Technologies, Grand Island, NY, USA) medium supplemented with 10% inactivated FBS (Gibco Life

Technologies) and penicillin (100 U/ml) / streptomycin (100 μg/ml) solution (Gibco Life Technologies) according to the manufacturer´s recommendations. Cells were maintained at 37˚C in a humidified 5% $CO_2$ atmosphere.

## Isolation of primary human lung fibroblasts

Adult human lung fibroblasts were obtained from lung biopsies of IPF patients who underwent surgical biopsy for the diagnosis of the disease (histologically confirmed usual interstitial pneumonia) [2]. The tissues were collected after obtaining written informed consent, as approved by and Ethics Committee of Bellvitge Hospital (CEIC, ref. PR202/08). The harvested lung tissue samples were maintained in a solution containing DMEM high Glucose with L Glutamine (Gibco Life Technologies) medium, HEPES (Sigma-Aldrich, St Louis, MO, USA) and insulin human transferrin and sodium selenite (ITS) (Sigma-Aldrich) until processing. Then, samples were cut into small pieces, and placed into six well plates (Nunc Thermo Scientific, Waltham, MA, USA) with growth medium; DMEM supplemented with 10% inactivated FBS (Gibco Life Technologies), penicillin (100 U/ml)/streptomycin (100 μg/ml) solution (Gibco Life Technologies) and 25 μg/ml amphotericin B (Sigma-Aldrich). Cells were cultured at 37˚C in a humidified atmosphere of 5% $CO_2$. Spindle-like primary fibroblasts started to grow out from tissue samples on day 2 to 3. Outgrowth of fibroblasts took 1 to 2 weeks. Tissue samples were then removed by aspiration, and cells were allowed to reach confluence. Fibroblasts at confluence were expanded by trypsinization and passaged every 4 to 5 days at 1:4 ratio. Cells between passages 4 and 7 were used in this study.

## Telomere length of IPF patients

IPF patients were divided into two groups according to the telomere length (TL). TL analysis was performed using DNA samples isolated from mouth epithelial cells and peripheral blood mononuclear cells (PBMC). Oral swabs (Isohelix, SK-2S, Cell Projects Ltd), previously validated in normal and with other telomeropathies subjects, were used for collection of cheek epithelial cells and DNA was extracted using a commercial kit (Isohelix, Cell Projects Ltd) [29–31]. DNA was obtained from PBMC, and the relative TL was assessed by quantitative polymerase chain reaction (qPCR) as previously described [32], and then was confirmed by Southern blot. The qPCR determines the ratio of telomere (T) repeat copy number to single-copy (S) gene (36B4) copy number (T/S ratio) in experimental samples, as compared with a reference DNA sample. This methodology was also validated for the TL measurement in buccal cells. As TL changes with age, a z-score value was obtained to allow the comparisons between individuals of different ages. The z-score compared the T/S value in each individual with the age-matched mean and SD of the values obtained in the controls (individual's value–population mean/population SD, age-matched population of within 9 years on average). The z-score below the 25[th] percentile of a normal distribution was considered telomere shortening. Severe TL reduction was identified when z-score was below the 1[st] percentile. Telomere shortening was also measured from blood DNA of each patient by Southern blot analysis of telomere restriction fragment (TRF) (TeloTAGGG Telomere Length Assay, Roche), which was considered the gold standard [32, 33].

## Human lung cell cultures treated with AT-MSC under TGF-β conditions

Human lung primary fibroblasts and A549 cell line were cultured in six well plates (Nunc Thermo Scientific) in the approprinated medium with 10% inactivated FBS; when cells reached 80% confluence the medium was changed to 2% inactivated FBS. Cells were stimulated with activated TGF-β (5 ng/ml) (R&D Systems Minneapolis, MN, USA) in the presence of several

doses of MP from AT-MSC (10,000; 50,000; and 100,000 MP/cell) during 48 h. After the incubation period, cells and supernatants were collected, then separated by centrifugation, and frozen at -80˚C for further analysis.

## Toxicity assay

Cell viability was evaluated using a commercial colorimetric assay (Quick Cell Proliferation Assay Kit II. MBL, International Corporation, Woburn, MA, USA) according to the recommended protocol. Cells ($5 \times 10^4$/well) were cultured in a 96-well microtiter plate (Nunc Thermo Scientific) and treated with different cell:MP ratio (1:10,000; 1:50,000 and 1:100,000) in a final volume of 100 μl/well of 2% FBS culture medium in triplicates for 48 h. Then, 10 μl/well of WST reagent was added and plates were incubated for 4 h at 37˚C in standard culture conditions. After shaking the plates for 1 min, the absorbance was computed at a wavelength of 420 nm in each well using a microplate reader (Thermo Scientific) with 650 nm of reference wavelength. The amount of the dye generated by activity of dehydrogenase is directly proportional to the number of living cells.

## Uptake of MP by A549 cells and fibroblasts

AT-MSC were labeled with red fluorescent PKH-26 dye according to the manufacturer's protocol (Sigma-Aldrich) prior to generation of MP, enabling the generation of fluorescent MP (PKH-MP). A549 and lung fibroblasts were plated at a density of $2 \times 10^5$ cells/well on a 12 well plate. Three ratios of PKH-MP, (1:10,000; 1:50,000 and 1:100,000) were added to the cultures for 24 h and 48 h. Then, the same protocol was performed by adding TGF-β (5 ng/mL) and the three ratios of PKH-MP for 48h. The uptake of MP by the cells was quantified by flow cytometry. The data were analyzed using Kaluza Software (Beckman Coulter).

For confocal microscopy images, Carboxyfluorescein succinimidyl ester (CFSE) was used to stain the A549 cells. The surface membrane of the fibroblasts was labeled with anti-CD44 and anti-CD90 antibodies both conjugated in APC. The nuclei of both types of cells were stained with 10μM Hoechst 33342 (Sigma). PKH-MP uptake by cells was imaged by a Leica TCS SP5 confocal microscope (Leica Microsystems B.V., Science Park Eindhoven, Netherlands), equipped with Leica Application Suite Advanced Fluorescence (LAS AF) software, DPSS 561 nm lasers, using a 60 X (1.4 NA oil) objective. Microscope images were processed using ImageJ 1.48 (National Institutes of Health, Washington, USA).

## RNA extraction and real-time polymerase chain reaction (RT-qPCR)

Total RNA was isolated from cultured cells using the Qiagen RNeasy Mini Kit (Qiagen, Valencia, CA, USA) according to the manufacturer's recommendations. Samples were digested with DNase I (Qiagen) to remove contaminating genomic DNA. One μg of RNA was reverse-transcribed using the iScript cDNA synthesis kit (Bio Rad) with oligo deoxythymidine and random hexamer primers. The reverse transcriptase reaction proceeded in a total volume of 20 μl in a conventional thermal cycler (Bio-Rad) at 25˚C for 5 min, followed by 30 min at 42˚C and 5 min at 85˚C. Reaction volumes of 20 μl were placed in 384-well optical reaction plates with adhesive covers (ABI Prism™ Applied Biosystems, Foster City, CA, USA) using SYBR Green PCR Master Mix and specific sequence primers (Sigma). The relative gene expression (RGE) of each targeted gene was normalized by subtracting the corresponding housekeeping genes (β-actin, HPRT and RNA18s) threshold cycle (Ct) value using the comparative Ct method (ΔΔCt methods).

## Western blot assay

Cells were lysed in Radio-Immunoprecipitation Assay (RIPA) Buffer (25 mM Tris-HCl, 150 mM NaCl, 1% NP-40, 0.1% sodium deoxycholate SDS) containing 1:100 phenylmethylsulfonyl fluoride and phosphatase inhibitors (Sigma). Prepared samples were heated at 100˚C for 4 min; for each sample the same amount of total protein (10 μg) was added to a well of 4–15% mini-protean TGX precast gels polyacrylamide gel (Bio-Rad Hercules, CA, USA) and resolved by SDS-PAGE. The separated proteins were transferred to a nitrocellulose membrane (Bio-Rad). The membranes were blocked for 1 h in Tris-buffered saline (10 mM Tris-HCl pH7.5 and 0.15 M NaCl) containing 0.1% (v/v) Tween 20 and 5% (w/v) bovine serum albumin (BSA) (Sigma-Aldrich), and then probed at room temperature (RT) for 1 h with primary antibodies against human EDA-fibronectin (diluted 1:400, Abcam). Immunoreactive bands were detected with IgG horseradish peroxidase-conjugated secondary antibodies (anti-mouse diluted 1:1,000) (Dako, Glostrup, Denmark) and visualized by enhance chemiluminescence detection reagents ECL Western blotting kit (Bio-Rad) according to the manufacturer´s instructions in a luminescent image analyzer (Chemi Doc Bio-Rad) and were then scanned for densitometry analysis (Image Lab Bio-Rad).

## ELISA

Levels of Tenascin-c and MMP-1 were measured in supernatants samples by Quantitative sandwich enzyme immunoassay for human MMP-1 (R&D Systems) and Human Tenascin-c large (FNIII-C) assay (IBL) according the manufacturer´s protocol.

## Statistical analysis

All results were expressed as mean ± SEM of independent experiments. Statistical analysis was performed using Graph Pad Prism 5.01 (Graph Pad Software, San Diego, CA, USA). Data were analyzed by one-way ANOVA, with post hoc testing using Bonferroni multiple comparison test and the non-parametric Mann Whitney test. A two-tailed P value of less than 0.05 was considered statistically significant.

# Results

## Patient's characteristics

The included IPF patients were diagnosed following the 2011 European Respiratory Society (ERS)/American Thoracic Society (ATS)/Japanese Respiratory Society/Latin American Thoracic Society Guidelines [2]. The final diagnosis was discussed and established by a multidisciplinary committee that includes expert ILD lung physicians, radiologists, and pathologists. Video-thoracic lung biopsy was required in all of them. Table 1 describes the characteristics of the patients at time of diagnosis and includes relevant clinical data such as gender, age,

**Table 1. Patient's demographics and characteristics.**

| | | | | | TELOMERE | LENGTH | |
| | | | | | T/S RATIO | Z-SCORE | |
| PATIENT | AGE AT DIAGNOSIS | RELATIVE WITH IPF | FVC% PREDICTED | DLCO% PREDICTED | T/S RATIO | Z-SCORE | PERCENTILE |
|---|---|---|---|---|---|---|---|
| #1 | 64 | YES | 77.6 | 60.6 | 0.728 | -1.824 | <1% |
| #2 | 70 | YES | 83.3 | 71.3 | 0.896 | -2.251 | <1% |
| #3 | 73 | YES | 74 | 42 | 0.890 | -2.272 | <1% |
| #4 | 66 | NO | 71.2 | 64.4 | 1.224 | -0.515 | 50–25% |
| #5 | 77 | NO | 66.1 | 83.1 | 1.418 | -0.126 | 50–25% |
| #6 | 73 | YES | 75.9 | 48 | 1.415 | -0.140 | 50–25% |

smoking history (packs per year), family history, and percentage of predicted value of lung function (FVC and DLco) according to reference values described by the ATS/ERS guidelines: standardisation of lung function testing [34, 35]. As telomere length is an important risk factor for disease development and poor outcome, we divided the obtained lung fibroblasts for cell culture experiments into two groups depending on the telomere length of the patients. Thus, they were classified into fibroblasts from patients with normal telomere length compared to age-matched controls, above the 25th percentile, designated as FN, and those with very short telomere length, below the first age-adjusted percentile, designed as FS. The mean percent predicted forced vital capacity (FVC%) of short telomere group was 78.3 ± 4.3%, and mean percent predicted carbon monoxide diffusing capacity (DLco%) was 57.9 ± 14.8%. For the normal telomere length group, the mean of FVC% was 71 ± 4.9 and the mean of DLco% was 65.1 ± 17.5. No respiratory physiological significant differences were found between patients with normal or shorten telomeres.

## Characterization of MP generated from AT-MSC

Cryoelectron microscopy analysis showed that all MP have a round shaped appearance with a discernible lipid bilayer (Fig 1A). The average size frequency of MP was 133.1± 22.1 nm (Fig 1B) and the average number of particles generated from each MSC was $1.3 \times 10^5 \pm 3.2 \times 10^4$ measured by NanoSight. The frequency of particles larger than 200 nm (cut-off pore size) was lower than 0.5 ± 0.3% in the samples indicating homogeneity in size of the MP population.

From the 2,998 proteins detected by mass spectrometry, five clusters were involved in the regulation of the ECM organization pathway. According to a classification based on the protein class, we observed protein modifying enzymes (metalloproteinase (MMP)-2 and 14) that participate in the degradation of the ECM; and protein binding activity modulator (Bone marrow stromal antigen 2 (BST2), Reversion-inducing cysteine-rich protein with Kazal motifs (RECK), and metalloproteinase inhibitor-3 (TIMP3)) that regulate the activity of both MMP-2, and 14 (Fig 1C).

To evaluate whether these proteins maintain their functional activity, and were not degraded during the generation of the MP, we performed a fluorimetric assay. MMP activity was observed in a dose-dependent manner. The MMP activity of MP was totally suppressed by adding the MMP inhibitor GM6001 during 15 min. Non-activated MMP were used as a background control for the assay (Fig 1D).

## MP do not induce toxicity on A549 cells and human lung fibroblasts

The potential toxicity of MP on A549 and lung fibroblasts from FS and FN IPF patients were determined by measuring the viability of these cells. Three cell:MP ratios (1:10,000; 1:50,000 and 1:100,000) were assayed for 48 h. Control cells were treated with the same volume of PBS. None of the tested ratios showed a significant decrease in the viability of A549 (Fig 2A), FN (Fig 2B) and FS (Fig 2C) compared to the PBS control.

## MP are effectively taken up by A549 cells and human lung fibroblasts

Two experimental conditions were carried out to analyze the MP uptake by A549. First, cells were cultured in presence of three ratios of fluorescent PKH-MP (cell:MP ratio 1:10,000; 1:50,000 and 1:100,000) during 24 and 48 h. The ratio of 10,000 MP per cell showed 45.1 ± 1.2 percentage of PKH26 positive A549 at 24 h and 52.9 ± 10.5 at 48 h. Probably, at the ratio of 10,000 MP per cell, the A549 cells did not accumulate enough PKH-MP inside to pass the fluorescent threshold and to become a clear positive population in the flow cytometry. However, at doses of 50,000 and 100,000 MP per cell, the percentage of PKH-MP positive cells reached the

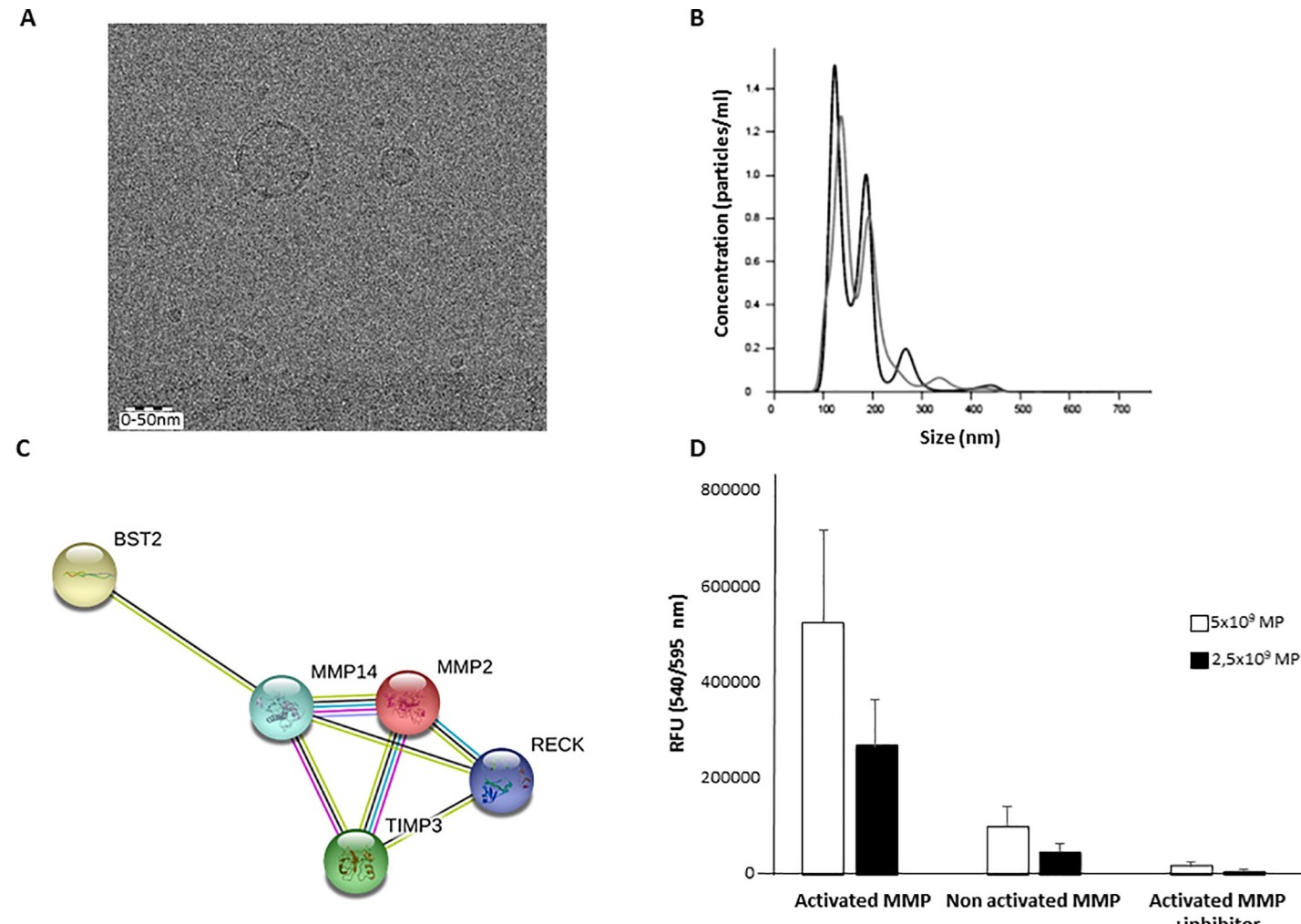

**Fig 1. Characterization of physical and molecular properties of MP from AT-MSC.** (A) Cryo-electron microscopy images of MP. MP show a spherical shape and a discernible lipid bilayer. (B) A representative profile of the nanoparticle tracking analysis (NTA) of MP. MP are visualized by light scattering using a light microscope. A video is taken, and the NTA software tracks the brownian motion of individual MP and calculates their size and total concentration (particles/ml). A graph was generated by plotting the distribution in size of the MP against the concentration. (C) STRING analysis (http://www.string-db.org) derived protein-protein interaction networks for 5 proteins that are involved in regulation of the extracellular matrix organization. The network nodes represent proteins. Edges represent protein-protein associations. The associations are meant to be specific and meaningful with associated proteins jointly contributing to a shared function. This does not necessarily mean that the proteins are physically binding each other. (D) MMP activity was measured in MP using a fluorimetric assay. MMP activity was activated with APMA and inhibited with GM6001. Results are expressed as mean ± SD.

100% of positive signal for both 24 and 48 h time points (Fig 3A). These results indicated that all cells were able to take up enough MP to be positive in the flow cytometry whether the doses were 50,000 or 100,000 MP per cell.

Second, A549 cells under TGF-β stimulation showed similar efficacy in the MP uptake compared to the non-stimulated condition (Fig 3B). Confocal microscope images were taken and, as shown in Fig 3C, MP (red dots) (ratio 1:50,000) were visualized in the cytoplasm (green color) but not in the nuclei (blue color) of the A549 cells after 48h of incubation. A representative orthogonal analysis is shown in Fig 3D to emphasize the internalization of MP.

Following the same experimental design as described above, lung fibroblasts from IPF patients also showed a high uptake efficacy of MP in non-stimulated condition (Fig 4A) with non-significant difference between 24 or 48 h of incubation time at doses of 1:50,000 and

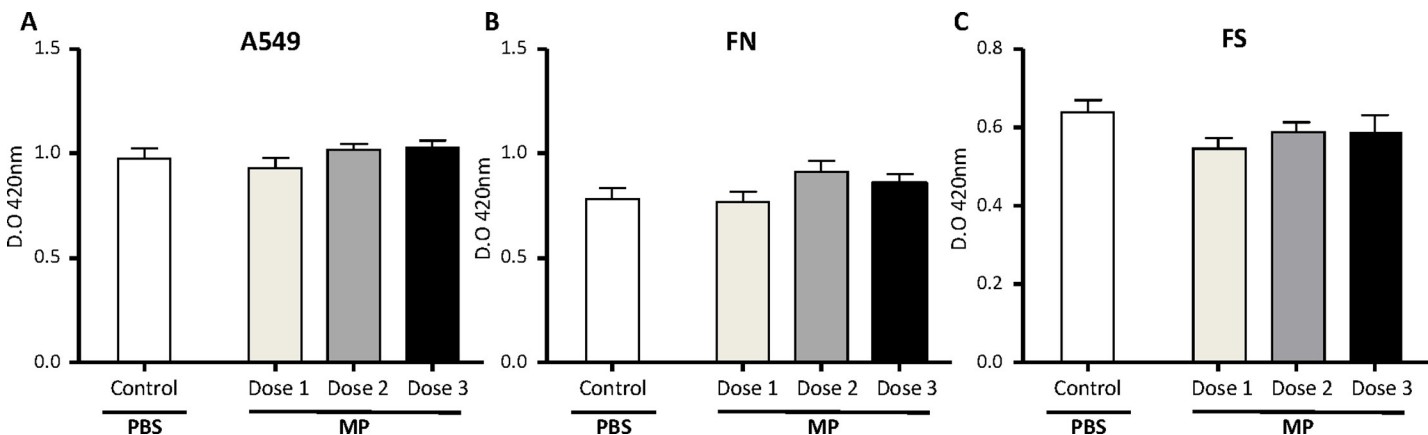

**Fig 2. Effect of MP in the viability of human lung cells.** Bronchial epithelial cells (A549) (A) and lung fibroblasts from normal (FN) telomere length patients (B) and short telomere length (FS) patients (C) were stimulated with different doses of MP for 48h. No significant toxicity was observed for any of the doses assayed; 1:10,000 ratio cells:MP (lighter grey bars), 1:50,000 ratio cell:MP (grey bars) and 1:100,000 ratio cell:MP (black bar) compared to PBS treatment (white bars). Bars represent the mean values ± SEM of optical density at 420 nm in three independent experiments.

1:100,000 MP per cell. Furthermore, TGF-β stimulation did not modify the uptake efficacy of MP by the cells (Fig 4B). Fig 4C shows the confocal images analysis at 48 h of incubation with TGF-β and MP at 1:50,000 dose and Fig 4D the internalization the MP by orthogonal analysis.

Taking together, the data from the toxicity assay, flow cytometry and confocal images, a dose of 50,000 MP per single cell was finally chosen for the following experiments.

## MP treatment reduces TGF-β-induced pro-fibrotic markers in A549 cells

A549 cell line was incubated for 48 h with MP from AT-MSC (1:50,000 cell:MP) and stimulated simultaneously with TGF-β (5 ng/ml). The modulation of genes involved in pro-(Fig 5A) and anti-fibrotic (Fig 5B) processes by MP was analyzed. As shown in Fig 5A TGF-β increased gene expression of the pro-fibrotic markers Collagen type I (Col 1A1), Collagen type III (Col 3A1), Fibronectin, and Plasminogen activator inhibitor 1 (PAI-1) with respect to the non-stimulated A549 culture (Control). MP treatment in TGF-β stimulated cells resulted in a significant gene expression decrease of these markers with respect to TGF-β control. Moreover, MP alone did not induce any pro-fibrotic response in non-stimulated cells resulting in a similar gene expression profile as control cells.

In Fig 5B, the anti-fibrotic markers (TERT, E-cadherin and MMP-7) were downregulated by TGF-β. When the gene expression of E-cadherin was studied, we did not find a recovery by MP of the inhibition provoked by TGF-β; which suggests that the epithelial mesenchymal transition (EMT) pathway activated by TGF-β was not reversible by the action of the MP. However, TGF-β no-stimulated cells treated with MP were able to express significant higher levels of E-cadherin compared to control. MMP-7 expression levels in TGF-β non-stimulated cells were decreased as compared to control, no significant results were found when cells were treated with MP in combination with TGF-β. Finally, when A549 cells were treated with MP, a significant increase in the gene expression of the telomerase component as compared to untreated cells was observed. However, MP were not able to recover the decrease of TERT gene expression induced by TGF-β at the control level.

## MP treatment decreased TGF-β fibrotic markers in human lung fibroblast

Following the same protocol as explained for epithelial cells, primary lung fibroblasts from patients with normal (FN) and short (FS) telomere length were analyzed under TGF-β with

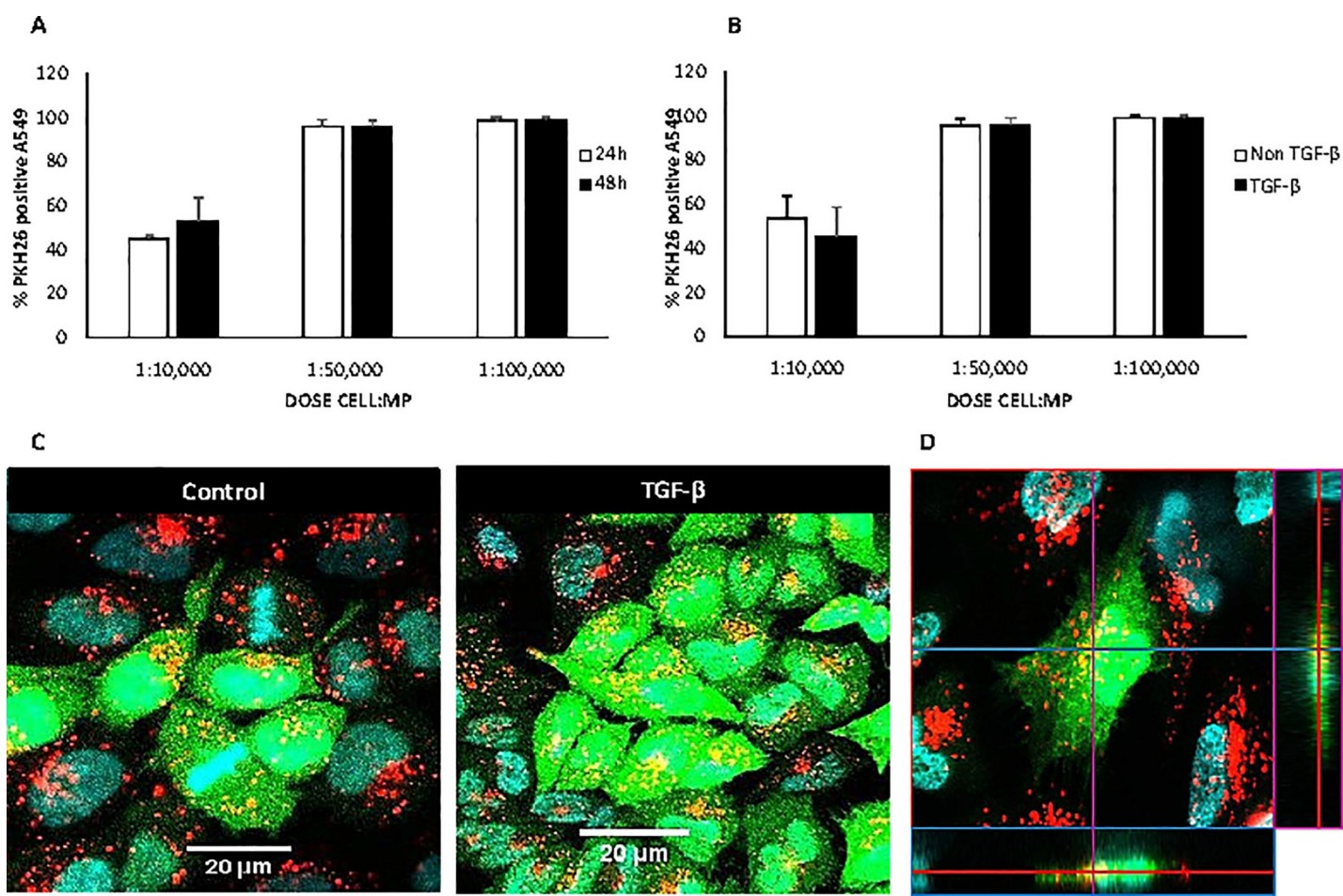

**Fig 3. Uptake analysis of MP by A549 cells.** AT-MSC were labeled with PKH-26 before generation of MP (PKH-MP). PKH-MP were added to A549 cells (cell:MP ratios 1:10,000; 1:50,000 and 1:100,000) and incubated for (A) 24 and 48 h. Uptake of PKH-MP by A549 cells was quantified using flow cytometry. Uptake is indicated by PKH-MP positive cells. (B) 48 h under TGF-β stimulation. (C) Representative confocal microscopy analysis of PKH-MP uptake by A549 cells at time point 48 h and with or without TGF-β stimulation. Staining for PKH-MP (red), cytoplasm (green), and Hoechst 33342 nuclei (blue) showed that PKH-MP are internalized by A549 cells.(D) Orthogonal view from different planes of the confocal microscope images to analyze the particle uptake. Co-localization of green fluorescent CFSE and red PKH-MP is observed in the cytoplasm. Levels of significance: * p < 0.05 respect to the ratio 1:50,000, and 1:100,000 cell:MP.

MP stimulation. TGF-β induced up-regulation of all the pro-fibrotic markers in lung fibro-blasts analyzed by gene and protein expression compared to the TGF-β non-stimulated control (Fig 6). Interestingly, fibroblast from FN and FS IPF patients had different responses in the modulation of the pro-fibrotic gene expression markers after TGF-β stimulation. Comparing the gene expression profile between FN and FS, three different responses were obtained. Fig 6A shows a similar pattern of the gene expression by the two types of fibroblasts. TGF-β stimulation provoked an increase of the pro-fibrotic markers COL 1A1, Tenascin-c, and PAI-1, that was reduced when cells were treated with MP. Moreover, there was a decrease in the anti-fibrotic protein, MMP-1, in cells stimulated with TGF-β that was reverted in MP treatment conditions. For the pro-fibrotic markers α-SMA, PD-L1, and Fibronectin, FN fibroblasts showed a higher modulatory response after MP treatment when compared to FS cultures. For these proteins, it seems that FS fibroblasts were not able to inhibit the TGF-β increase when cells were treated with MP (Fig 6B). Finally, there were two pro-fibrotic markers COL 3A1 and TGF-β, which showed a statistically significant decrease of gene activation by TGF-β after MP treatment in FS but not in FN fibroblasts (Fig 6C).

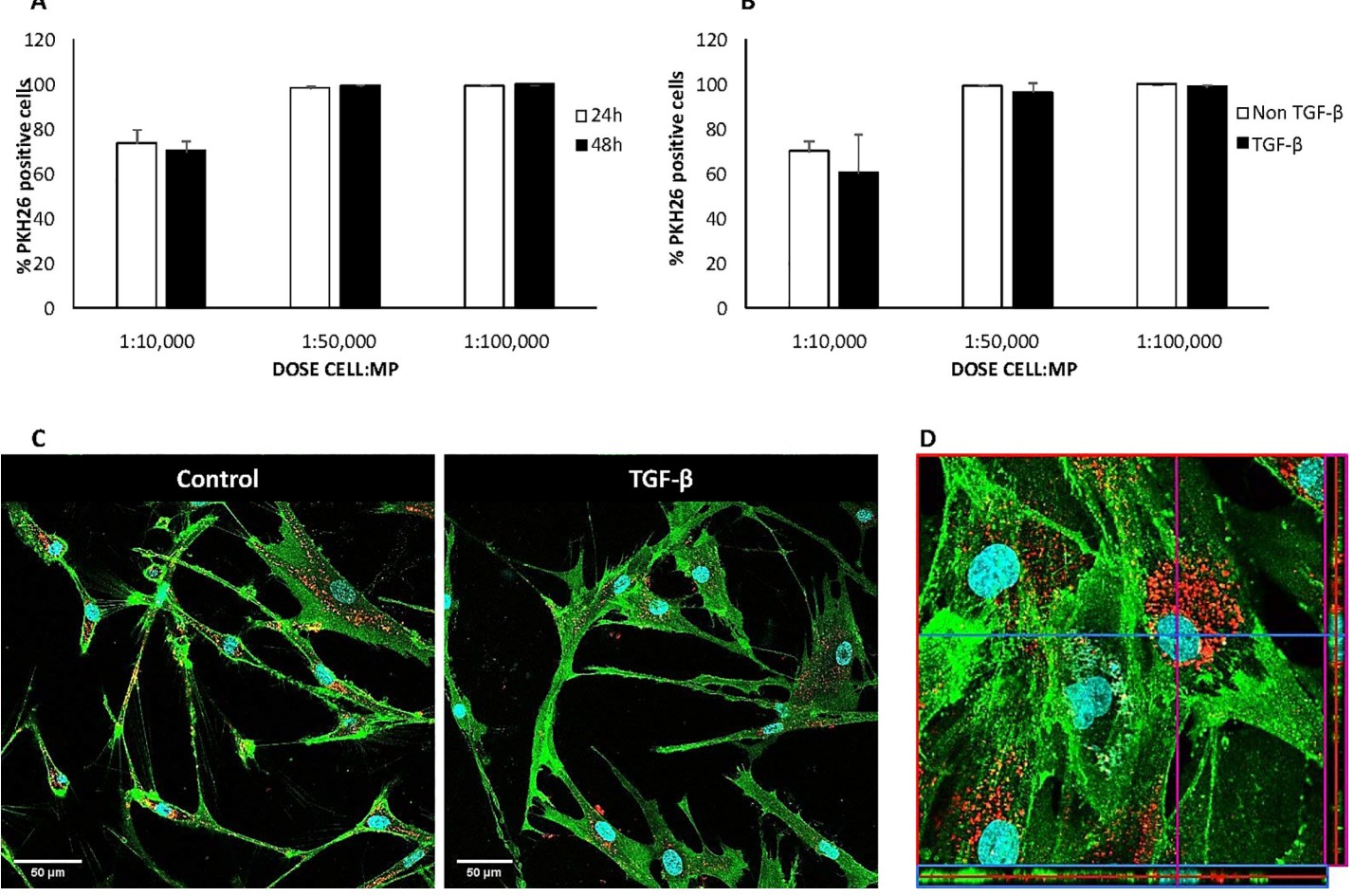

**Fig 4. Uptake of PKH-MP by human lung fibroblasts.** PKH-MP uptake by fibroblast was analyzed by flow cytometry at the time points (A) 24 and 48 h. (B) 48 h under TGF-β stimulation. (C) *In vitro* internalization of PKH-MP into human fibroblasts at 48 h time point and with or without TGF-β stimulation. PKH-MP are shown as red color, cell membrane (green), and Hoechst 33342 nuclei (blue). (D) Representative orthogonal analysis. Levels of significance: * p < 0.05 respect to the ratio 1:50,000, and 1:100,000 cell:MP.

The gene expression of vimentin was also analyzed, but MP did not induce any significant difference when compared to TGF-β stimulated cells (data not shown).

Some of the markers analyzed by qPCR were then studied for protein expression. The protein concentration of Tenascin-c (Fig 7A) and MMP-1 (Fig 7B) was analyzed in the supernatant of the fibroblast cultures, and levels of Fibronectin protein were analyzed in cell lysates (Fig 7C and 7D). The levels of Tenascin-c and Fibronectin were reduced by MP treatment under TGF-β stimulation in FN and FS, and the concentration of anti-fibrotic MMP-1 protein was increased in both types of fibroblasts. These results supported the gene expression data.

## Discussion

Despite constant efforts, many therapeutic approaches for IPF patients failed until anti-fibrotic therapy altered the management of this devastating disease. Currently, pirfenidone and nintedanib are the only approved drugs that slow down IPF progression [11–13], but there is no definitive cure for avoiding the lethality of the disease. Furthermore, the patient response to anti-fibrotic treatment is heterogeneous [9], stressing the need to establish novel therapeutic

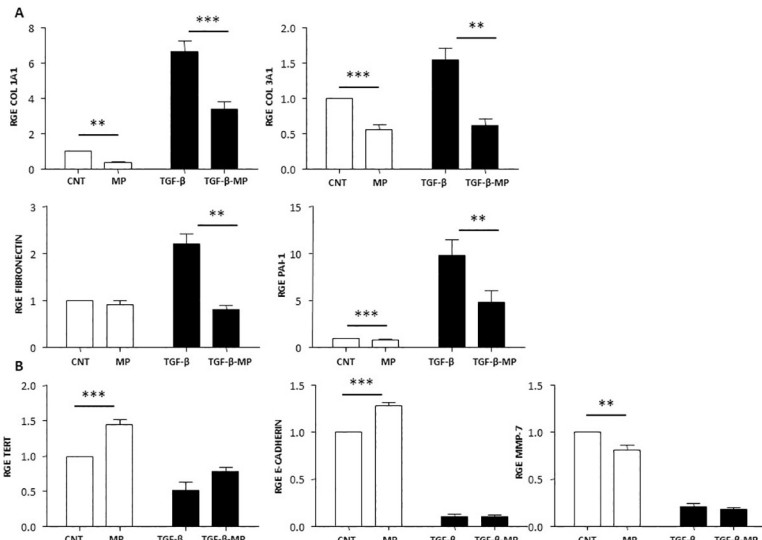

**Fig 5. MP effect in gene expression in bronchial epithelial cells.** A549 cells were cultured with 1:50,000 cell:MP ratio and TGF-β (5 ng/ml) for 48 h. Gene expression of different markers involved in ECM remodeling pathways was analyzed by qPCR. (A) The pro-fibrotic markers Col1A1, Col3A1, fibronectin and PAI-1. (B) The anti-fibrotic molecules TERT, E-cadherin, and MMP7. Bars represent mean values ± SEM fold changes expressed as relative gene expression (RGE) of three different experiments. Levels of significance: $^*$ p < 0.05, $^{**}$ p < 0.01; $^{***}$ p < 0.001. $^*$ indicates the comparison between control and MP treated cells or TGF-β stimulated cells compared to the combination of TGF-β and MP.

approaches. The present *in vitro* study demonstrates that MP from AT-MSC can ameliorate TGF-β induced fibrotic responses in human lung epithelial cells and fibroblasts by reducing the gene expression of proteins involved in fibrogenesis.

A growing body of experimental and preclinical studies on lung pathologies, including IPF, supports the efficacy of the administration of MSC from different sources such as bone marrow [14], placenta [19], lung tissue [15], and adipose tissue [18]. These studies show that MSC administration, independently of the cell source, is safe and have a beneficial effect in fibrotic lung diseases. The authors reported that MSC should be infused early in the disease course [36], during active inflammation and before significant fibrotic changes [37]; otherwise, MSC might acquire a myofibroblast phenotype and contributed to fibrotic changes during the fibrotic phase of the disease [37].

Several studies described the comparison of MSC properties derived from different tissue origin [38–40]. However, there is still a concern about which tissue provides the most appropriate type of MSC for therapy. We have observed that AT-MSC had the best proliferation rate in culture, and same immunomodulatory properties when comparing to other MSC sources such as umbilical cord and bone marrow [41]. Mediating the regenerative properties of AT-MSC through MP could offer several physiological advantages such as reducing the risk of cellular dysfunction because MP lack the capacity to proliferate. Furthermore, MP are unable to modify their behavior in response to the environment, making MP therapy better controllable than MSC therapy. The methodology described herein allowed obtaining MP with intact transmembrane protein as observed by the detection of the metalloproteinase enzyme activity, thereby exploiting some of the natural regenerative properties of AT-MSC.

Another beneficial characteristic of MP is that they were effectively taken up in large quantities by alveolar epithelial cells (A549) and lung fibroblasts from IPF patients without inducing cytotoxic effects. As shown in the confocal microscopy images, there was the internalization of

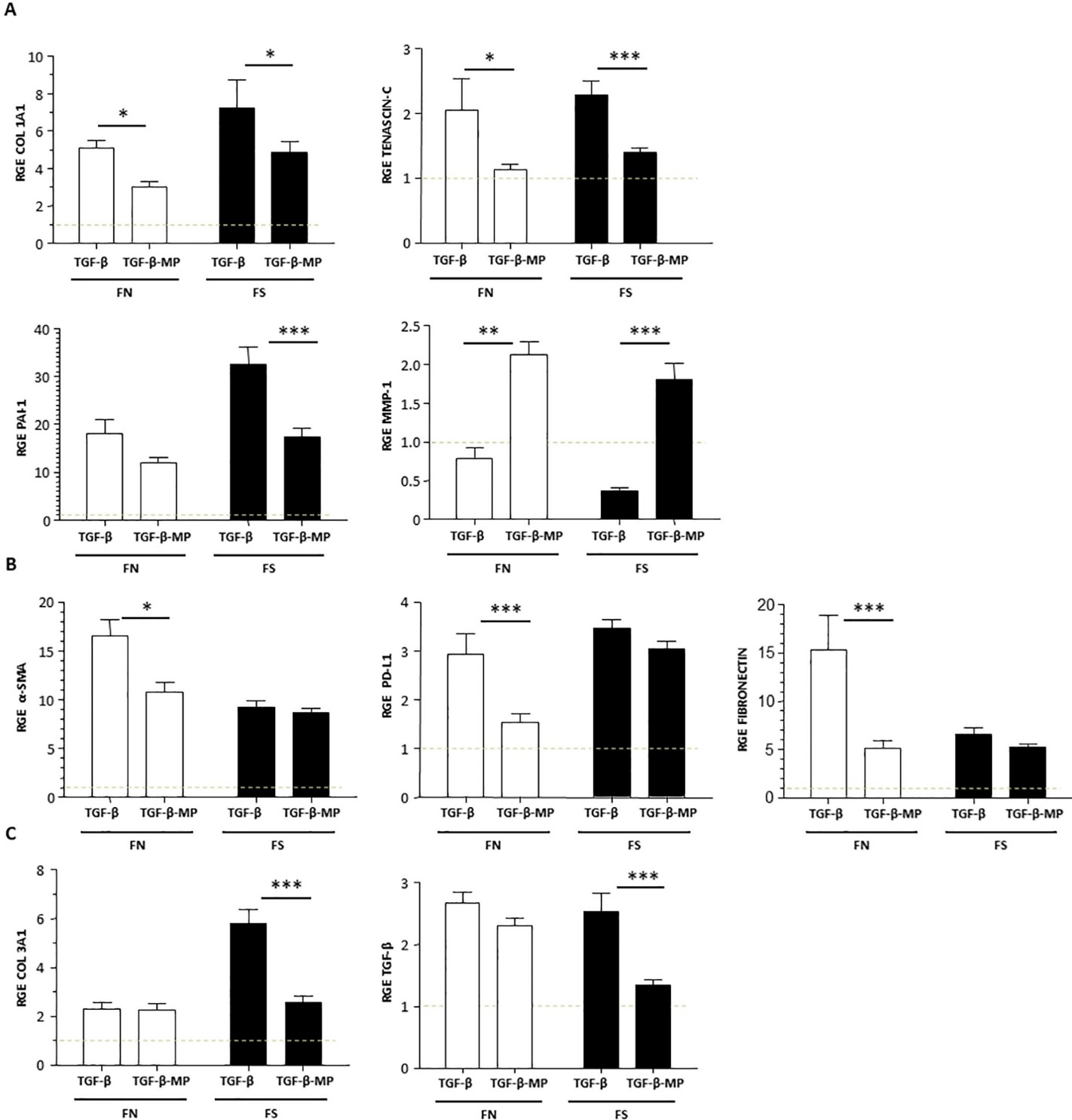

**Fig 6. MP effect on gene expression in human lung fibroblasts.** Lung fibroblasts from patients with normal (FN) and short (FS) telomere length were stimulated with 1:50,000 cell:MP ratio and with or without TGF-β (5 ng/ml) for 48 h. Gene expression was assayed by qPCR. (A) Comparison of two types of fibroblasts showing a similar profile in the modulation of pro-fibrotic markers; COL 1A1, tenascin-c, and PAI-1 and the anti-fibrotic MMP-1 gene. (B) Higher response in gene expression of α-SMA, PDL-1 and fibronectin markers in FN than FS cultured cells. (C) Differential modulation in gene expression of COL 3A1 and TGF-β showing FN culture lower response compared to FS. Grey dashed line represents the normalization based on the gene expression level of untreated control according to the ΔΔCt method. Chars represent mean values ± SEM fold changes expressed as relative gene expression (RGE) of nine different experiments. Levels of significance: * p < 0.05, ** p < 0.01; *** p < 0.001. * indicates the comparison between TGF-β stimulated cells compared to TGF-β and MP.

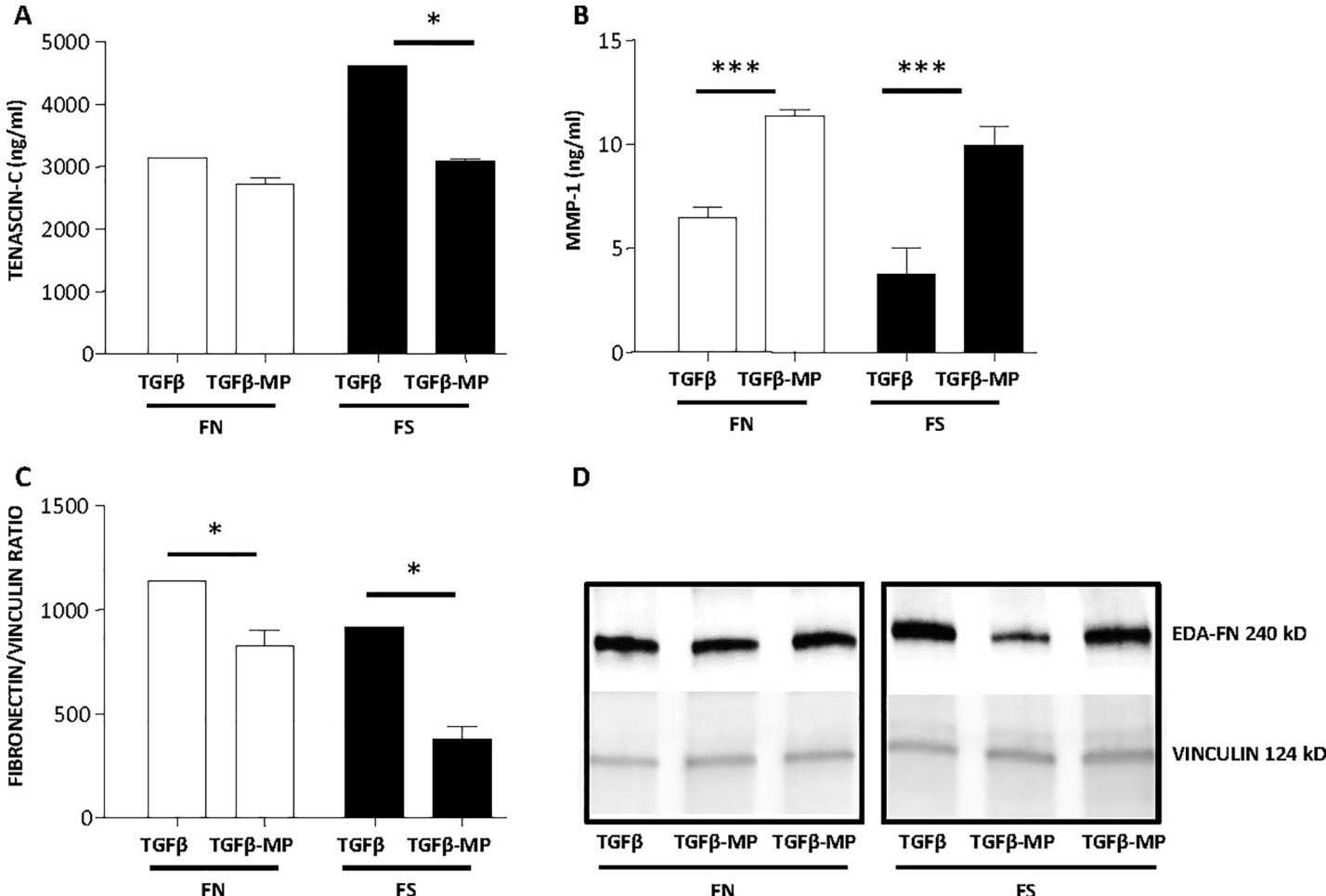

**Fig 7. MP effect in protein levels.** Fibroblasts from IPF patients with normal (FN) and short (FS) telomere length were stimulated with TGF-β (5 ng/ml) and 1:50,000 cell:MP ratio for 48 h. Protein levels in supernatants were analyzed by ELISA for tenascin-c (A), and MMP-1 (B). Protein expression of fibronectin in cell extracts was analyzed by Western blot, bars of fibronectin values normalized with vinculin expression (C), representative Western blot images of one TGF-β stimulated sample and two different TGF-β-MP treated samples (D). Chars represent mean values ± SEM of nine independent experiments. Levels of significance: * p < 0.05, ** p < 0.01 *** p < 0.001. * indicates the comparison between TGF-β stimulated cells compared to TGF-β and MP.

MP into lung cells. Previously, we have observed that macrophages and endothelial cells appear to take up MP by a variety of endocytic pathways, including clathrin-dependent endocytosis, and clathrin-independent pathways such as caveolin-mediated uptake, and phagocytosis [42]. These mechanisms of internalization, shared with EV [43], support the idea that MP may be used as well as delivery vehicle for lung cells-targeting drugs with inherent anti-fibrotic effects.

We used the A549 cell line to study the effect of MP on cell-released proteins. Although the A549 cell line is from a cancer patient, it has been broadly used to model alveolar type II epithelial cells behavior [44, 45]. MP decreased the gene expression of profibrotic proteins such as Col 1A1, Fibronectin and PAI-1 in A549 cells. However, MP did not achieve a recovery of E-cadherin expression after its inhibition by TGF-β, indicating that the EMT pathway is not totally abolished with MP treatment. Several groups described similar findings on A549 cells using pirfenidone, suggesting that the E-cadherin inhibition by TGF-β might not be reversible [46, 47]. An important result in our study was the increase in TERT expression in A549 cells after MP treatment in non TGF-β stimulated condition.

It has been described that the dysfunction of telomerase associates telomere attrition and, as a result, accelerate aging in these cells [48–50]. By enhancing TERT expression in epithelial cells, MP might have a protective role against senescence and potentially favor the recovery of the alveolus architecture.

Fibroblasts from IPF patients with different telomere length were used to analyze the anti-fibrotic role of MP in TGF-β stimulated cultures. It has been described that telomere length is correlated with the clinical outcome of IPF patients [5]. We found different effects of MP in fibroblasts of patients with long and short telomeres. Thus, some of the markers decreased in FN upon MP treatment; such as α-SMA, PD-L1 and Fibronectin, but others were significantly reduced in FS; Col 3A1 and TGF-β. However, for most of the important pro-fibrotic markers both types of fibroblasts had the same inhibitory response after MP treatment Col 1A1, Tenascin-c, PAI-1 and an activation of anti-fibrotic MMP-1 gene expression. It has been described that aging is a key factor in IPF, and lung fibroblasts from old mice exhibit a fibrogenic phenotype that leads to increased susceptibility to fibrosis after injury [16, 51, 52]. The differential response shown by the fibroblasts in this study might be due to different cell behavior depending on biological age, including the patient telomere attrition. Little is known about the different IPF fibroblast function depending on aging hallmarks, and the present study stratified the lung fibroblastic response depending on the patient telomere length. Thus, one might hypothesize that FS have a more severe fibrotic response after TGF-β. MP treatment was able to inhibit the TGF-β increased gene expression by decreasing the levels of the pro-fibrotic factors involved in fibrogenesis. MP may be a suitable option for attenuating the fibrotic response in IPF patients with normal and short telomere length. Further studies are needed to better characterize the molecular implication of MP in the inhibition of TGF-β mediated fibrosis.

These results are in agreement with a previous study of our group that showed a decrease in pro-fibrotic markers in lung cells that were stimulated with a combination of pirfenidone and rapamycin in a similar TGF-β induced *in vitro* model of fibrosis [53]. This suggests that MP treatment have a comparable modulatory effect of the main ECM proteins involved in the fibrotic pathway to these conventional drugs.

Although our results seem promising, this *in vitro* study is a proof-of-concept about the potential effect of MP in the lung cell pro-fibrotic response and present several limitations. First, *in vitro* studies are needed to improve our understanding about the mechanisms of action of the MP. However, we might assume that the transmembrane proteins present in the MP have a role in the modulation of many cell pathways by comparing with the findings reported in several EV studies [54–56]. Second, *in vivo* studies should be performed to further support the role of MP as a potential anti-fibrotic therapy.

In conclusion, MP represent a nano-size version of MSC membranes that conserve some of the therapeutic properties attributed to the living MSC. Our data demonstrates that MP target lung cells, via which they may have a broad anti-fibrotic effect.

## Supporting information

**S1 Fig.**
(TIF)

## Author Contributions

**Conceptualization:** Ana Merino, Ana Montes-Worboys.

**Formal analysis:** Ana Merino, Elena G. Arias-Salgado, Sander S. Korevaar, Ana Montes-Worboys.

**Funding acquisition:** Martin J. Hoogduijn, Maria Molina-Molina.

**Investigation:** Ana Merino, Ana Montes-Worboys.

**Methodology:** Ana Merino, Elena G. Arias-Salgado, Ana Montes-Worboys.

**Project administration:** Ana Merino, Ana Montes-Worboys.

**Software:** Sander S. Korevaar.

**Supervision:** Ana Merino, Ana Montes-Worboys.

**Validation:** Martin J. Hoogduijn, Maria Molina-Molina, Carla C. Baan.

**Writing – original draft:** Ana Merino, Ana Montes-Worboys.

**Writing – review & editing:** Ana Merino, Martin J. Hoogduijn, Maria Molina-Molina, Carla C. Baan, Ana Montes-Worboys.

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
