## [Editor Report · Decision Letter 0]

19 Oct 2020

PONE-D-20-29201

Membrane particles from mesenchymal stromal cells reduce the expression of fibrotic markers on pulmonary cells

PLOS ONE

Dear Dr. Ana Montes Worboys,

Thank you for submitting your manuscript to PLOS ONE. After careful consideration, we feel that it has merit but does not fully meet PLOS ONE’s publication criteria as it currently stands. Therefore, we invite you to submit a revised version of the manuscript that addresses the points raised during the review process.

We look forward to receiving your revised manuscript.

Kind regards,

Sujeong Jang

Academic Editor

PLOS ONE

Journal Requirements:

You have to check the English proof.

"This collaboration project is co-funded by the PPP Allowance made available by Health

498 Holland, Top Sector Life Sciences & Health, of the Dutch Ministry of Economic affairs

499 to stimulate public-private partnerships, and by the Institute of Health Carlos III (ISCIII,

500 PI18/00367), co-funded by FEDER funds/European Regional Development Fund

501 (ERDF) – “a Way to Build Europe” and Biomedical National Research Network

502 CIBER."

i) We note that you have provided funding information that is not currently declared in your Funding Statement. However, funding information should not appear in the Acknowledgments section or other areas of your manuscript. We will only publish funding information present in the Funding Statement section of the online submission form.

ii) Please remove any funding-related text from the manuscript and let us know how you would like to update your Funding Statement. Currently, your Funding Statement reads as follows:

" The authors received no specific funding for this work.".

6. We noticed you have some minor occurrence of overlapping text with the following previous publication(s), which needs to be addressed:

- Gonçalves, F.D.C. (Fabiany Da C.), Luk, F, Korevaar, S.S, Bouzid, R. (Rachid), Paz, A.H. (Ana H.), López-Iglesias, C. (Carmen), … Hoogduijn, M.J. (2017). Membrane particles generated from mesenchymal stromal cells modulate immune responses by selective targeting of pro-inflammatory monocytes. Scientific Reports, 7(1). doi:10.1038/s41598-017-12121-z

The text that needs to be addressed involves the last paragraph of the discussion (conclusion).

In your revision ensure you cite all your sources (including your own works), and quote or rephrase any duplicated text outside the methods section. Further consideration is dependent on these concerns being addressed.

Additional Editor Comments (if provided):

They try to isolate and uptake MP into the cells, and follow them for using therapeutic tools.

There are several comments for update the data and protocols.

1. Following your methods for isolation of MP, how many volume of MP (concentration) did you get? You mentioned the particle shape, size, and etc., but not concentration.

2. Why did you use the H2O for MP isolation? There are a lot of protocols to isolate nanoparticles, using commercial kit, solution, ultra-centrifuge, irradiation, etc. But you used H2O in this work. Do you have any references for it?

3. At line 357, you mentioned 'The results indicated that the uptake was dose-dependent.'. You have to compare the different dose results, not time. In Fig.3A, you have to compare and mark dose-dependently difference as paired t-test.

4. How long did you uptake the particles into the cells in Fig3C and Fig4C?

5. As Fig3C, the cells look like different, such as shape and amount. Did you think that the cells are differentiated into other cells? As Fig4C, the cells morphology is quiet different between control vs. TGF-b.
---

## [Author Response · Author response to Decision Letter 0]

5 Nov 2020

RESPONSE TO REVIEWERS

As per suggestion of the Editor, we checked the style requirements of the PLOS ONE journal, and we changed the Manuscript accordingly. 

All necessary patients’ data are described in the Manuscript to replicate our study findings. We do not have any legal or ethical restrictions for publication on sharing data publicly. To clarify this point, we changed the paragraph by deleting the availability of data upon request. 

We provided the original blot figure to note there were no manipulation of the images. 

The corresponding Authors have validated the ORCID ID in Editorial Manager.

"This collaboration project is co-funded by the PPP Allowance made available by Health

498 Holland, Top Sector Life Sciences & Health, of the Dutch Ministry of Economic affairs

499 to stimulate public-private partnerships, and by the Institute of Health Carlos III (ISCIII,

500 PI18/00367), co-funded by FEDER funds/European Regional Development Fund

501 (ERDF) – “a Way to Build Europe” and Biomedical National Research Network

502 CIBER."

i) We note that you have provided funding information that is not currently declared in your Funding Statement. However, funding information should not appear in the Acknowledgments section or other areas of your manuscript. We will only publish funding information present in the Funding Statement section of the online submission form.

ii) Please remove any funding-related text from the manuscript and let us know how you would like to update your Funding Statement. Currently, your Funding Statement reads as follows:

" The authors received no specific funding for this work.".

Following the Editor advice, we changed the funding paragraph in the Manuscript, and included the correct information in the cover letter to be changed by the Journal.

“The collaboration project is co-funded by the PPP Allowance made available by Health~Holland, Top Sector Life Sciences & Health, to stimulate public-private partnerships, and by the Institute of Health Carlos III (ISCIII, PI18/00367), co-funded by FEDER funds/European Regional Development Fund (ERDF) – “a Way to Build Europe” and Biomedical National Research Network CIBER.”

6. We noticed you have some minor occurrence of overlapping text with the following previous publication(s), which needs to be addressed:

- Gonçalves, F.D.C. (Fabiany Da C.), Luk, F, Korevaar, S.S, Bouzid, R. (Rachid), Paz, A.H. (Ana H.), López-Iglesias, C. (Carmen), … Hoogduijn, M.J. (2017). Membrane particles generated from mesenchymal stromal cells modulate immune responses by selective targeting of pro-inflammatory monocytes. Scientific Reports, 7(1). doi:10.1038/s41598-017-12121-z

The text that needs to be addressed involves the last paragraph of the discussion (conclusion).

In your revision ensure you cite all your sources (including your own works), and quote or rephrase any duplicated text outside the methods section. Further consideration is dependent on these concerns being addressed.

Following the suggestion of the Editor, we have modified the conclusion of the manuscript to avoid any overlap with previous work and cited the appropriate reference. 

Additional Editor Comments (if provided):

They try to isolate and uptake MP into the cells, and follow them for using therapeutic tools.

There are several comments for update the data and protocols.

1. Following your methods for isolation of MP, how many volume of MP (concentration) did you get? You mentioned the particle shape, size, and etc., but not concentration.

As per suggestion, we have included the required information in the manuscript, results section (Characterization of MP generated from AT-MSC). 

2. Why did you use the H2O for MP isolation? There are a lot of protocols to isolate nanoparticles, using commercial kit, solution, ultra-centrifuge, irradiation, etc. But you used H2O in this work. Do you have any references for it?

We agree with the Editor´s comment in the confusion of isolation method of MP could generate. We clarified in the text, that we did not isolate extracellular vesicles, but generated membrane particles from MSCs. These MP are artificially generated, so the isolation method is completely different than the protocols for EV collection. The first step of our protocol is to induce an osmotic lysis by adding H2O to the cells. Then, the nuclei of the cells are flashed out and we can collect the cell membranes without DNA contamination. This protocol was described in our first article (reference 24 of this manuscript). The protocol of MP generation has been updated providing more details in the revised version of this manuscript (methods: Generation and Characterization of Membrane Particles from AT-MSC). 

3. At line 357, you mentioned 'The results indicated that the uptake was dose-dependent.'. You have to compare the different dose results, not time. In Fig.3A, you have to compare and mark dose-dependently difference as paired t-test.

Following the suggestion of the Editor, the results were modified in the Manuscript, section “MP are efficiently taken up by A549 cells and human lung fibroblasts”, the statistic was added to the graph and legend.

4. How long did you uptake the particles into the cells in Fig3C and Fig4C?

The cells were incubated 48h because is also the time needed for the TGFb to stimulate the cells. The incubation time was also added in the result section.

5. As Fig3C, the cells look like different, such as shape and amount. Did you think that the cells are differentiated into other cells? As Fig4C, the cells morphology is quiet different between control vs. TGF-b.

We agree with the Editor that TGF-b provoke the differentiation of both epithelial and fibroblast cells in vitro. This is supporting by plenty of literature that report the transformation of epithelial into mesenchymal cells (EMT) or fibroblasts into myofibroblast after TGF-b treatment. In fact, we used the TGF-b as a fibrotic model for its transforming properties. We described this in the Introduction section.

---

## [Decision Letter · Decision Letter 1]

6 Jan 2021

PONE-D-20-29201R1

Membrane particles from mesenchymal stromal cells reduce the expression of fibrotic markers on pulmonary cells

PLOS ONE

Dear Dr. Worboys,

Thank you for submitting your manuscript to PLOS ONE. After careful consideration, we feel that it has merit but does not fully meet PLOS ONE’s publication criteria as it currently stands. Therefore, we invite you to submit a revised version of the manuscript that addresses the points raised during the review process.

ACADEMIC EDITOR: 

Major concerns have been raised over the content of your manuscript, and I have now recommended it for major revision. 

We look forward to receiving your revised manuscript.

Kind regards,

Sujeong Jang

Academic Editor

PLOS ONE

Reviewers' comments:

Reviewer's Responses to Questions

**Comments to the Author**

1. If the authors have adequately addressed your comments raised in a previous round of review and you feel that this manuscript is now acceptable for publication, you may indicate that here to bypass the “Comments to the Author” section, enter your conflict of interest statement in the “Confidential to Editor” section, and submit your "Accept" recommendation.

Reviewer #1: (No Response)

Reviewer #2: All comments have been addressed

Reviewer #3: (No Response)

2. Is the manuscript technically sound, and do the data support the conclusions?

Reviewer #1: No

Reviewer #2: Partly

Reviewer #3: Partly

3. Has the statistical analysis been performed appropriately and rigorously? 

Reviewer #1: No

Reviewer #2: Yes

Reviewer #3: Yes

4. Have the authors made all data underlying the findings in their manuscript fully available?

Reviewer #1: Yes

Reviewer #2: Yes

Reviewer #3: Yes

5. Is the manuscript presented in an intelligible fashion and written in standard English?

Reviewer #1: Yes

Reviewer #2: Yes

Reviewer #3: No

6. Review Comments to the Author

Reviewer #1: I still have some concerns abouth this manuscript.

1. Panel labelling should be included in Figure 7. Such as Figure 7a, b, c and d. In addition, the unit of bottom left is wired, which starts from 1200? And the Western blot at bottom right is not well labeling. Dose TGF-MP indicate two lanes?

2. In Figure 6A and 6B, FN in TGF vs TGF-MP seems significant difference to me.

3. western blot should be "W"estern blot in all content.

Reviewer #2: The revised manuscript has addressed previous review critique well. However, this reviewer found four major deficiencies in the current form of submission. First, the usefulness of membrane particles (MPs) prepared from AT mesenchymal stromal cells in inhibiting fibrotic activities was carried out only at the in vitro cell line studies. The impact of MPs on the treatment of fibrosis in vivo was not demonstrated. Thus, the usefulness of the MPs in vivo is not clear. Second, there is no mechanistic experiment demonstrated in this paper on how MPs in the inhibition of the fibrotic activity. In the absence of the mechanistic study, off-targeted effects may occur. Third, the experiment designs are poorly carried out. MPs were used at one dose, there is no dose-dependent data to support the claim. In addition, what is the approach to standardize the treatment, based on protein level or others? It is not clear if MPs prepared from non-AT MSC will have the similar effects or not. If this is the case, then the control experiment should use the MPs from non-AT MSC sources instead of using PBS treatment as the control. Lastly, A549 cancer cell line is not an appropriate epithelial cell type to represent alveolar epithelial cells. Th cancer cell line is transformed and their differentiated nature is not the same as the alveolar epithelial cells.

Typo: Middle in Abstract: A459 is not A549.

Reviewer #3: The paper is technically sound. However there are some errors in presentation of the data, and some things should be reworded so as not to make conclusions that possibly go beyond the results.

The paper raises many questions such as the mechanism underlying the effect, the fate of the MPs (are they internalized to lysosomes and degraded?), whether the effect depends on the cell of origin for the MPs, how much protein is in the MPs and how much is membrane-bound, etc. However these questions can be considered beyond the scope of the paper.

37. “Efficiently” is not the correct word. Efficiency is a measure of how much output is obtained from a given input – eg, how much power is produced from a given amount of fuel burned. If anything, this process seems inefficient in that tens of thousands of MPs per cell are needed to show that there is uptake. See other uses of the word eg 349.

38. “inhibited…” I think is the wrong word. The MPs reduced expression of these proteins, but did not inhibit them, ie, did not inhibit their function.

44. “promising…” I think the conclusion that the findings indicate that MPs are a “promising therapy” is not supported by the findings, although admittedly “promising” can be interpreted with some latitude. The conclusions should be restricted to the data, in my opinion, which do not address therapy. That said, no reader is likely to be misled by this, so this is a style suggestion.

78. “no methodology to obtain … a protein free final isolated product.” Line 70 suggests that a protein-free product would not work. I do not understand why a protein-free product would be a goal.

118. “A population of MP, homogeneous in size was obtained by extruding the plasma membranes through polycarbonate membrane filters (Merck, KGaA, 120 Darmstadt, Germany) from a pore diameter of 800 nm to 200 nm.” Please explain this better. This step was not in ref 24. Did you use filters with 800nm pores, then 200nm? Were there any sizes in between? Is it known if this step simply removes particles above the size limit, or are MPs above the size limit fragmented/extruded through the pores (the word “extruding” makes it sounds like larger particles might be forced through the pores)?

119. Do you know that these are (just) plasma membranes? Could the material be a mixture of various types of membranes (plasma, vesicles, ER, Golgi, etc)? If you do not know that they are ony plasma membranes then would suggest referring to them more generically as membranes or cell membranes.

169. The numbers reported are not concentrations, they are just total number of particles. “Briefly” is confusing because it sounds like you are saying that the incubation was for a brief period.

170. The paper seems to claim that (intact?) MPs have MMP activity – or at least it is ambiguous in this regard. (“The MMP activity of MP was totally suppressed by adding the MMP inhibitor GM6001”). If the intact MPs have MMP activity, is it because the assay substrate is able to diffuse into the MP (in which case the MPs might not be able to use their MMPs to degrade a larger protein substrate located outside the MPs), or because the MMP activity is on the surface of the MP? The MMPs are activated by “1M APMA”; does this high concentration of APMA disrupt the MPs, releasing internal MMPs? (Did you mean 1mM instead? 1M is a very high concentration, and I don't think AMPA is soluble at that concentration.) If the point is just to confirm the mass spec results by showing that there is some kind of MMP enzyme activity, make the wording more definitive so that a reader wont think you are claiming that the intact MPs have been proved to degrade a substrate like collagen.

222. “below the 10th and 1st percentiles”? As written, this just means below the 1st percentile.

313. “predicted pulmonary lung function test” needs to be rewritten. The lung function was measured, not predicted. “Pulmonary lung” is redundant.

318. “designed” means “designated”?

328. “Double layer MP were also visualized in the samples.” This was also shown by cryo-EM? Please give at least a rough indication of how many particles were double layer compared to just a single lipid bilayer.

329. "The average peak size frequency of MP was 133.1± 22.1 nm". The number shown is a size; it is unclear what "peak size frequency" means.

375. “As shown in Fig A” should be “Fig 5A”?

472. “As all the patients included in our study had the same IPF diagnosis and similar chronological age, the only different factor between FN and FS were the length of their telomeres.” This cannot be true.

476. “blocking the pro-fibrotic factors” should be rephrased. See 38, above.

609. I do not understand figure B or the legend: “A graph was generated by plotting the distribution in size of the MP against the concentration of MP per ml.” The figure appears to plot size (not distribution in size) against intensity (not concentration). The legend does appear to describe Fig 2a in ref 24. What do the squares mean (they have different sizes and shadings)? Please explain this properly.

610. The Y axis is in units of AU, not in MP/ml. I understand that AU is likely proportional to MP/ml, but the text should refer to the units shown.

625. A reader might wonder how many PKH-MPs do there need to be in one cell before it passes the threshold of positivity in the flow cytometry experiment? In the 1:10,000 part, for example, roughly half the cells are not “positive” but does that mean that they have MPs, but below the limit of detection, or that they have no MPs? Maybe a histogram would be a better way to present the flow data. I assume there would be a curve with one peak, rather than a bimodal curve with positive and negative cells. The histogram would make clear that you had to choose a cutoff threshold. This is a suggestion, because I don't think the paper is harmed by leaving as is.

626. The figure says 1:10, 1:50 etc instead of 1:10k, 1:50k etc. (It is not clear, just from looking at the figure, what is meant by the x103 written to the side – ie, one might reasonably think that 1:10 x 103 is equal to 103:10, rather than 1:104).

626. In the figure 3 legend, what is “DOSIS”? Same for Figure 4.

Figures 3C, 4C. These are confocal images, but not enough information is given to convince me that the MPs are within the cells, as opposed to sitting on the outside. If you are claiming that the MPs are truly within the cells, please mention supporting information, such as Z-stack, otherwise restate to say something like the imaging shows the MPs associated with the cells.

There are many minor grammar mistakes that should be corrected.

Examples:

#A therapy that avoids MSC side effects of transformation would be an alternative for [to] the use of living cells.

#with the formation of fibroblast and myofibroblasts [myofibroblast] foci [1] where TGF-β play [plays] a crucial

#pirfenidone [11] and nintedanib [12] are the only treatment [treatments]

#Finally, the low yield of EV originated by the cells make [makes] difficult an [a] scalable production

7. PLOS authors have the option to publish the peer review history of their article (what does this mean?). If published, this will include your full peer review and any attached files.

Reviewer #1: No

Reviewer #2: No

Reviewer #3: No

---

## [Author Response · Author response to Decision Letter 1]

27 Jan 2021

Response to the reviewers

Reviewer #1: I still have some concerns about this manuscript.

1. Panel labelling should be included in Figure 7. Such as Figure 7a, b, c and d. In addition, the unit of bottom left is wired, which starts from 1200? And the Western blot at bottom right is not well labeling. Dose TGF-MP indicate two lanes?

We thank the Reviewer´s comments and we´ve changed the Figure 7 accordingly. In the Western blot panel, the two lanes under the same bar represent two different representative samples treated with TGF-MP. We have modified the legend figure to clarify this point.

2. In Figure 6A and 6B, FN in TGF vs TGF-MP seems significant difference to me.

Although we agreed with the Reviewer, after checking again the statistical analysis, we did not find a p value < 0.05 to be significant.

3. western blot should be "W"estern blot in all content.

We corrected the term Western blot along the revised Manuscript as per suggestion.

Reviewer #2: The revised manuscript has addressed previous review critique well. However, this reviewer found four major deficiencies in the current form of submission. First, the usefulness of membrane particles (MPs) prepared from AT mesenchymal stromal cells in inhibiting fibrotic activities was carried out only at the in vitro cell line studies. The impact of MPs on the treatment of fibrosis in vivo was not demonstrated. Thus, the usefulness of the MPs in vivo is not clear. 

We thank the Reviewer’s comment about the first version of the revised Manuscript. We agree that this work has a main limitation due to the lack of the animal model; however, the aim of the research was to establish a proof of concept about the role of the Membrane Particles derived from mesenchymal stem cells in lung fibrosis. We have added in the discussion a paragraph including the limitations of the study to avoid any misinterpretation of the conclusions. 

Second, there is no mechanistic experiment demonstrated in this paper 1on how MPs in the inhibition of the fibrotic activity. In the absence of the mechanistic study, off-targeted effects may occur. 

In agreement with the Reviewer, the mechanism of action of MP is a very interesting topic to study. MP are not a soluble factor that activate the different signaling pathways involved in the fibrotic process depending on the interaction with a specific receptor. Lung cells uptake MP by endocytosis and phagocytosis in a continuous manner up to 48 h (Fig 3 and 4). We might hypothesize the activation or down-regulation of relevant pathways by analyzing the modulation of the relevant mediators implicated in pulmonary fibrosis. We’ve included this limitation in the Discussion section.

Third, the experiment designs are poorly carried out. MPs were used at one dose, there is no dose-dependent data to support the claim.

Before the final dose was chosen for the following stimulation experiments, we performed a study with three different ratios of MP (10,000; 50,000 and 100,000 MP/cell). The ratio 10,000 MP/cell didn´t show a significant effect in our readout. For the ratios 50,000 and 100,000; almost 100% of cells resulted positive for the MP uptake and the fibrotic markers expression were similar. However, at dose of 100,000 MP/cell we found many particles in the supernatants that they were not uptake by the cells due to the overload of the cells. We did not include those data in the Manuscript to simplify the Result section.

In addition, what is the approach to standardize the treatment, based on protein level or others? 

The standardization of the treatment for the experiments is based on the NanoSight. This technique give us the number of particles/ml that we have after the generation of the MP. In addition, the MP generation is always carried out in the same way, same number of cells per flask and same volume and concentration of the different reagents. We considered that protein level was not a good standardization because MP are more than a pool of proteins.

It is not clear if MPs prepared from non-AT MSC will have the similar effects or not. If this is the case, then the control experiment should use the MPs from non-AT MSC sources instead of using PBS treatment as the control.

We agree with the Reviewer, we do not know if the effect of the MP is specific of the cell source and to date the effects of MSC are specific of these type of cells or a simple fibroblast has the same properties. In our experience (Curr Opin Organ Transplant. 2014;19(1):41-6.), we have indicated that AT-MSC have the best proliferation rate in culture and same immunomodulatory properties when comparing to another MSC sources. Furthermore, they are easier to collect by less invasive method than bone marrow-MSC, from patients to undergo kidney donation in our Institution.

Lastly, A549 cancer cell line is not an appropriate epithelial cell type to represent alveolar epithelial cells. Th cancer cell line is transformed and their differentiated nature is not the same as the alveolar epithelial cells.

We are aware that bronchoalveolar A549 cell line is from a cancer patient. However, the A549 cell line has been broadly used to model alveolar type II epithelial cells behavior, not only for the difficulty in the isolation and culture of the primary cells, but to obtain enough quantity of protein and RNA to explore our objectives (Oncotarget, 2020 11:1306-1320) (Cell Death Discovery, 2019 5:146). In addition, we did not observe any transformation of the control cell line (A549) during the culture period.

Typo: Middle in Abstract: A459 is not A549.

We´ve corrected this mistake in the Abstract.

Reviewer #3: The paper is technically sound. However, there are some errors in presentation of the data, and some things should be reworded so as not to make conclusions that possibly go beyond the results. The paper raises many questions such as the mechanism underlying the effect, the fate of the MPs (are they internalized to lysosomes and degraded?), whether the effect depends on the cell of origin for the MPs, how much protein is in the MPs and how much is membrane-bound, etc. However these questions can be considered beyond the scope of the paper.

We agree that the raised Reviewer´s questions will be key for a better understanding of the MP mechanisms, and we are performing the studies to address these topics. In this new open research line, we already found that MP are internalized by endocytosis and phagocytosis, and they were observed in organelles such as Golgi, Endoplasmic reticulum and lysosomes (Basic Science in Transplantation Meeting, 2018). We have also analyzed, by mass spectrometry, the protein content of the MP and found proteins associated to cell membrane from several organelles, not only from plasma membrane (Manuscript in preparation). 

37. “Efficiently” is not the correct word. Efficiency is a measure of how much output is obtained from a given input – eg, how much power is produced from a given amount of fuel burned. If anything, this process seems inefficient in that tens of thousands of MPs per cell are needed to show that there is uptake. See other uses of the word eg 349.

In agreement with the Reviewer, we have changed the word efficiently for effectively, to mean that the amount of MP inside the cell is the optimal to induce an effect. 

38. “inhibited…” I think is the wrong word. The MPs reduced expression of these proteins, but did not inhibit them, ie, did not inhibit their function.

We have changed this word in the revised version.

44. “promising…” I think the conclusion that the findings indicate that MPs are a “promising therapy” is not supported by the findings, although admittedly “promising” can be interpreted with some latitude. The conclusions should be restricted to the data, in my opinion, which do not address therapy. That said, no reader is likely to be misled by this, so this is a style suggestion.

We agreed with the Reviewer and re-written the conclusion to avoid misinterpretation of the results. We have added a paragraph with the limitations of the study.

78. “no methodology to obtain … a protein free final isolated product.” Line 70 suggests that a protein-free product would not work. I do not understand why a protein-free product would be a goal.

We re-written this sentence in the Introduction to better explain the main disadvantage of the EV isolation protocol is that soluble proteins non-associated with EV would co-precipitated making difficult to distinguish whether the effect on the target cells is due to the EV themselves or by the contaminant proteins contained in the final sample. 

118. “A population of MP, homogeneous in size was obtained by extruding the plasma membranes through polycarbonate membrane filters (Merck, KGaA, 120 Darmstadt, Germany) from a pore diameter of 800 nm to 200 nm.” Please explain this better. This step was not in ref 24. Did you use filters with 800nm pores, then 200nm? Were there any sizes in between? Is it known if this step simply removes particles above the size limit, or are MPs above the size limit fragmented/extruded through the pores (the word “extruding” makes it sounds like larger particles might be forced through the pores)?

We have re-phrased this paragraph in the methodology to clarify the extrusion process. MP suspension is forced by pressure through a defined pore side filter to yield particles having a diameter near the last pore size. We used three different gradually size filters of 800 nm, 400 nm and 200 nm pores. 

119. Do you know that these are (just) plasma membranes? Could the material be a mixture of various types of membranes (plasma, vesicles, ER, Golgi, etc)? If you do not know that they are ony plasma membranes then would suggest referring to them more generically as membranes or cell membranes.

As we responded above, we found that MP contain proteins from other organelles not only from plasma membrane. We have changed plasma membrane by cell membrane in the revised Manuscript. 

169. The numbers reported are not concentrations, they are just total number of particles. “Briefly” is confusing because it sounds like you are saying that the incubation was for a brief period.

We have modified the text as per suggestion.

170. The paper seems to claim that (intact?) MPs have MMP activity – or at least it is ambiguous in this regard. (“The MMP activity of MP was totally suppressed by adding the MMP inhibitor GM6001”). If the intact MPs have MMP activity, is it because the assay substrate is able to diffuse into the MP (in which case the MPs might not be able to use their MMPs to degrade a larger protein substrate located outside the MPs), or because the MMP activity is on the surface of the MP? The MMPs are activated by “1M APMA”; does this high concentration of APMA disrupt the MPs, releasing internal MMPs? (Did you mean 1mM instead? 1M is a very high concentration, and I don't think AMPA is soluble at that concentration.) If the point is just to confirm the mass spec results by showing that there is some kind of MMP enzyme activity, make the wording more definitive so that a reader won’t think you are claiming that the intact MPs have been proved to degrade a substrate like collagen.

We agreed with the Reviewer that this paragraph may be confusing and it has been re-written in the revised version to clarify. The proteomic study showed MP have two different MMPs (2 and 14) (Fig. 1C). MMP-2 may be soluble protein and MMP-14 is a trans-membrane protein, as this kit measures all of the MMP activity, we might assume that MP have MMP activity both in the surface and inside them. 

The Reviewer is right, the AMPA working solution is 2mM, we´ve changed this in the text (1M is the stock solution). 

We agree with the Reviewer that we cannot conclude that MP are able to degrade ECM proteins but this was not the aim of this experiment. We intended to demonstrate that proteins, found in MP, were not degraded during their generation and they maintain their activity. This has been clarified in the revised Manuscript. 

222. “below the 10th and 1st percentiles”? As written, this just means below the 1st percentile.

The Reviewer´s comment is correct. We have changed the phrase for a better understanding.

313. “predicted pulmonary lung function test” needs to be rewritten. The lung function was measured, not predicted. “Pulmonary lung” is redundant.

We have changed this concept as per suggestion. 

318. “designed” means “designated”?

We corrected this term and changed with designated in the revised version.

328. “Double layer MP were also visualized in the samples.” This was also shown by cryo-EM? Please give at least a rough indication of how many particles were double layer compared to just a single lipid bilayer.

We thank the Reviewer´s comment and we have modified this sentence in the Manuscript to avoid the confusion. All the MP found by cryo-EM were lipid bilayer, we did not observed any single layer. 

329. "The average peak size frequency of MP was 133.1± 22.1 nm". The number shown is a size; it is unclear what "peak size frequency" means. 

We have corrected this sentence as per suggestion.

375. “As shown in Fig A” should be “Fig 5A”?

We have corrected this mistake in the revised version.

472. “As all the patients included in our study had the same IPF diagnosis and similar chronological age, the only different factor between FN and FS were the length of their telomeres.” This cannot be true.

We agreed with the Reviewer that this paragraph might sound confusing. We changed the text to avoid any misinterpretation, and we have suggested that one of the s for the differences found in the experiments might be due to the differences in the telomere length of the patients. 

476. “blocking the pro-fibrotic factors” should be rephrased. See 38, above.

We have revised these terms along the revised Manuscript.

609. I do not understand figure B or the legend: “A graph was generated by plotting the distribution in size of the MP against the concentration of MP per ml.” The figure appears to plot size (not distribution in size) against intensity (not concentration). The legend does appear to describe Fig 2a in ref 24. What do the squares mean (they have different sizes and shadings)? Please explain this properly.

We agree with the Reviewer, the graph represents intensity of the particles scattering against size. We have replaced this panel by another easier to follow in the figure 2B.

610. The Y axis is in units of AU, not in MP/ml. I understand that AU is likely proportional to MP/ml, but the text should refer to the units shown.

The revised version of the figure 2B has the new graph with the proper units.

625. A reader might wonder how many PKH-MPs do there need to be in one cell before it passes the threshold of positivity in the flow cytometry experiment? In the 1:10,000 part, for example, roughly half the cells are not “positive” but does that mean that they have MPs, but below the limit of detection, or that they have no MPs? Maybe a histogram would be a better way to present the flow data. I assume there would be a curve with one peak, rather than a bimodal curve with positive and negative cells. The histogram would make clear that you had to choose a cutoff threshold. This is a suggestion, because I don't think the paper is harmed by leaving as is.

The Reviewer´s comment is right. In the 1:10,000 ratio, the flow cytometry histogram shows one peak that overlaps 50% with the non-MP control. When we observed these cells in the confocal microscopy, all of them contain MP but seems that the fluorescent intensity is not strong enough to show a clear positive population in the flow cytometry due to the detection limit of the technique. When we incubated the lung cells at the ratio 1:50,000 and 1:100,000, we obtained a clear positive peak that does not overlap with the negative peak in the cytometer. We have added a sentence in the result section to clarify this point.

626. The figure says 1:10, 1:50 etc instead of 1:10k, 1:50k etc. (It is not clear, just from looking at the figure, what is meant by the x103 written to the side – ie, one might reasonably think that 1:10 x 103 is equal to 103:10, rather than 1:104).

We´ve modified the figure to better understand the meaning of the graphs. 

626. In the figure 3 legend, what is “DOSIS”? Same for Figure 4.

We´ve corrected the word dosis with the doses.

Figures 3C, 4C. These are confocal images, but not enough information is given to convince me that the MPs are within the cells, as opposed to sitting on the outside. If you are claiming that the MPs are truly within the cells, please mention supporting information, such as Z-stack, otherwise restate to say something like the imaging shows the MPs associated with the cells.

We have added a new panel (D) to the figures 3 and 4 to show the orthogonal view of the cells in order to emphasize the internalization of MP.

There are many minor grammar mistakes that should be corrected.

We appreciated the grammar correction of the Reviewer and we carefully checked the spelling for other mistakes in the revised Manuscript. 

Examples:

#A therapy that avoids MSC side effects of transformation would be an alternative for [to] the use of living cells.

#with the formation of fibroblast and myofibroblasts [myofibroblast] foci [1] where TGF-β play [plays] a crucial

#pirfenidone [11] and nintedanib [12] are the only treatment [treatments]

#Finally, the low yield of EV originated by the cells make [makes] difficult an [a] scalable production

7. PLOS authors have the option to publish the peer review history of their article (what does this mean?). If published, this will include your full peer review and any attached files.

Do you want your identity to be public for this peer review? For information about this choice, including consent withdrawal, please see our Privacy Policy.

Reviewer #1: No

Reviewer #2: No

Reviewer #3: No

---

## [Editor Report · Decision Letter 2]

16 Feb 2021

PONE-D-20-29201R2

Membrane particles from mesenchymal stromal cells reduce the expression of fibrotic markers on pulmonary cells

PLOS ONE

Dear Dr. Worboys,

Thank you for submitting your manuscript to PLOS ONE. After careful consideration, we feel that it has merit but does not fully meet PLOS ONE’s publication criteria as it currently stands. Therefore, we invite you to submit a revised version of the manuscript that addresses the points raised during the review process.

We look forward to receiving your revised manuscript.

Kind regards,

Sujeong Jang

Academic Editor

PLOS ONE

Additional Editor Comments (if provided):

In the revised manuscript the authors made only small improvements. The answers given by authors to reviewer’s comments are not convincing. Specific comments:

1. Following your reply for Figure 6A and 6B, you did not find any significant between the result. You have to describe the exact statical mean+/-SEM. After that, we will assume your results carefully.

2. Answering for reviewer 3 the authors wrote still ‘a potential promising therapy’ in abstract. The results are not supported as a promising therapy.

3. It is not enough to satisfy for reviewer 3 why a protein-free product would be a goal. If you have any reference, you describe it in the manuscript directly.

4. In materials and methods, you mentioned about the procedure about polycarbonate membrane filter. However, we cannot find any details in your manuscript. Even thought, you used three different size filters, we couldn’t understand well. Did you use them step by step or individually or respectively?

5. In results, what is the reference values in line #321 in revised manuscript?

6. Answering for reviewer 3’s comment, the author wrote “we have modified this sentence in the Manuscript to avoid the confusion. All the MP found by cryo-EM were lipid bilayer, we sis not observed any single layer” in 2nd sentence in Results ‘characterization of MP generated from AT-MSC’. However, you just removed the 2nd sentence without any modification. Is it your final answering?

7. Reviewer 3 mentioned that the author should explain the figure legend for understanding in figure 1B. However, author did not describe anything in here. Even though the reviewer wants to know what the concentration of MP per ml means, there is no description.

8. All limitations mentioned by reviewer should be at least shortly described in Discussion section. In the revised manuscript Discussion part was almost the same as in original submission although the reviewer asked for improvements. In addition, the author just omitted the description without addition.

---

## [Author Response · Author response to Decision Letter 2]

25 Feb 2021

Response to the Reviewers:

Additional Editor Comments (if provided): In the revised manuscript the authors made only small improvements. The answers given by authors to reviewer’s comments are not convincing. Specific comments: 

1. Following your reply for Figure 6A and 6B, you did not find any significant between the result. You have to describe the exact statical mean+/-SEM. After that, we will assume your results carefully.

After a deeper revision of the statistical analysis and discussion with the Statistician of our institution, the Reviewer was right and the results of figures 6A and 6B were significant. We run the Mann Whitney test, and obtained the following mean±SEM data for COL1A1 (6A): TGF-β (5.122 ± 0.3703) vs TGF-β-MP (3.013 ± 0.2569) and for alpha-sma (6B): TGF-β (16.51 ± 1.686) vs TGF-β-MP (10.83 ± 0.9152). In both cases the p value was <0.05.

We apologize for missing the correction in the last Revised version of the manuscript, we have modified accordingly the figure by adding the asterisk in the correct columns. 

We thank the Reviewer´s comment and appreciate the opportunity to correct this mistake. 

2. Answering for reviewer 3 the authors wrote still ‘a potential promising therapy’ in abstract. The results are not supported as a promising therapy.

As per suggestion, we have changed the conclusion paragraph in the Abstract section. 

3. It is not enough to satisfy for reviewer 3 why a protein-free product would be a goal. If you have any reference, you describe it in the manuscript directly.

As it is being described by the International Society of Extracellular Vesicles (J Extracell Vesicles. 2018;7(1):1535750), obtaining a pure preparation of EV is still a challenge. During the isolation protocol, soluble proteins co-precipitate with the EV being a confounding factor in the interpretation of the results. It is not possible to know whether the effect of the preparation is due to the EV or the contaminated proteins. This is the reason that a contaminant-protein-free product would be the ideal goal to better know the true effect of EV. 

In the MP generation protocol, we avoid this co-precipitation of proteins and other soluble factors by discarding the supernatant of the cells. 

We have added and cited the new reference in the Introduction section. 

4. In materials and methods, you mentioned about the procedure about polycarbonate membrane filter. However, we cannot find any details in your manuscript. Even thought, you used three different size filters, we couldn’t understand well. Did you use them step by step or individually or respectively?

We gradually pass through different pore size filters the MP, from bigger to smaller size (800-400-200 nm). The sequential use of these filters make it possible to obtain the final desired particle size at 200nm without blockade the filter. This is a method described for liposome preparation (Zhang H. Thin-Film Hydration Followed by Extrusion Method for Liposome Preparation. Methods Mol Biol. 2017;1522:17-22. doi: 10.1007/978-1-4939-6591-5_2. PMID: 27837527). 

We´ve re-written the methodology section to better clarify this point.

5. In results, what is the reference values in line #321 in revised manuscript?

The term reference values refers to predicted values of standardization of lung function testing. (Eur Respir J 2005; 26: 319–338 and Eur Respir J 2005; 26: 720–735).

We have added these references in the Revised Manuscript. 

6. Answering for reviewer 3’s comment, the author wrote “we have modified this sentence in the Manuscript to avoid the confusion. All the MP found by cryo-EM were lipid bilayer, we sis not observed any single layer” in 2nd sentence in Results ‘characterization of MP generated from AT-MSC’. However, you just removed the 2nd sentence without any modification. Is it your final answering?

We agree with the Reviewer that we just removed the second sentence because it was confusing but, in our opinion, there is nothing else to add. This sentence, in the previous version, made the reader think that we had both single double phospholipid layers in our sample, and we had only bilayer. 

7. Reviewer 3 mentioned that the author should explain the figure legend for understanding in figure 1B. However, author did not describe anything in here. Even though the reviewer wants to know what the concentration of MP per ml means, there is no description.

We have modified the legend in figure 1B according to the Reviewer´s comment.

8. All limitations mentioned by reviewer should be at least shortly described in Discussion section. In the revised manuscript Discussion part was almost the same as in original submission although the reviewer asked for improvements. In addition, the author just omitted the description without addition.

We thank the Reviewer´s comment and, as per suggestion, we have changed the Discussion section to improve the final version of the Manuscript by adding all the potential limitations and concerns raised by the Reviewers.

---

## [Editor Report · Decision Letter 3]

26 Feb 2021

Membrane particles from mesenchymal stromal cells reduce the expression of fibrotic markers on pulmonary cells

PONE-D-20-29201R3

Dear Dr. Worboys,

We’re pleased to inform you that your manuscript has been judged scientifically suitable for publication and will be formally accepted for publication once it meets all outstanding technical requirements.

Kind regards,

Sujeong Jang, PhD.

Academic Editor

PLOS ONE

Additional Editor Comments (optional):

The paper was improved according to the reviewer's suggestions.
---

## [Editor Report · Acceptance letter]

2 Mar 2021

PONE-D-20-29201R3 

Membrane particles from mesenchymal stromal cells reduce the expression of fibrotic markers on pulmonary cells 

Dear Dr. Montes-Worboys:

I'm pleased to inform you that your manuscript has been deemed suitable for publication in PLOS ONE. Congratulations! Your manuscript is now with our production department. 

Kind regards, 

on behalf of

Dr. Sujeong Jang 

Academic Editor

PLOS ONE